# One bout of neonatal inflammation impairs adult respiratory motor plasticity in male and female rats

Austin D Hocker[1], Sarah A Beyeler[1], Alyssa N Gardner[2], Stephen M Johnson[2], Jyoti J Watters[2], Adrianne G Huxtable[1]*

[1]Department of Human Physiology, University of Oregon, Eugene, United States; [2]Department of Comparative Biosciences, University of Wisconsin-Madison, Madison, United States

**Abstract** Neonatal inflammation is common and has lasting consequences for adult health. We investigated the lasting effects of a single bout of neonatal inflammation on adult respiratory control in the form of respiratory motor plasticity induced by acute intermittent hypoxia, which likely compensates and stabilizes breathing during injury or disease and has significant therapeutic potential. Lipopolysaccharide-induced inflammation at postnatal day four induced lasting impairments in two distinct pathways to adult respiratory plasticity in male and female rats. Despite a lack of adult pro-inflammatory gene expression or alterations in glial morphology, one mechanistic pathway to plasticity was restored by acute, adult anti-inflammatory treatment, suggesting ongoing inflammatory signaling after neonatal inflammation. An alternative pathway to plasticity was not restored by anti-inflammatory treatment, but was evoked by exogenous adenosine receptor agonism, suggesting upstream impairment, likely astrocytic-dependent. Thus, the respiratory control network is vulnerable to early-life inflammation, limiting respiratory compensation to adult disease or injury.
DOI: https://doi.org/10.7554/eLife.45399.001

*For correspondence: huxtable@uoregon.edu

Competing interests: The authors declare that no competing interests exist.

## Introduction

At birth, neonates transition from a sterile maternal environment into an environment filled with pathogens, microbes, and toxins and must simultaneously begin robust, rhythmic breathing. Respiratory problems represent a significant clinical problem for neonatologists (*Martin et al., 2012*), especially in preterm infants where breathing is unstable (*Poets and Southall, 1994*; *Poets et al., 1994*) and infections are common (*Stoll et al., 2002*; *Stoll et al., 2004*). Further, inflammation appears to augment respiratory dysfunction in neonates, whereby inflammation depresses hypoxic responses (*Olsson et al., 2003*; *Rourke et al., 2016*) and induces recurrent apneas (*Hofstetter et al., 2007*). Despite the prevalence of early life inflammation, little is known about the long-lasting consequences of neonatal inflammation on adult neurorespiratory control.

We are beginning to understand the potential for long-term consequences of early life inflammation in other physiological systems. Neonatal inflammation blunts adult immune function (*Bilbo et al., 2010*; *Mouihate et al., 2010*; *Spencer et al., 2011*), increases adult stress reactivity (*Shanks et al., 2000*; *Wang et al., 2013*; *Grace et al., 2014*), impairs adult learning and hippocampal plasticity (*Bilbo, 2005a*; *Bilbo et al., 2006*), increases the risk of neuropsychiatric disorders (*Rantakallio et al., 1997*; *Hornig et al., 1999*), and worsens age-related cognitive decline (*Bilbo, 2010*). Yet, we know very little about the long-term effects of neonatal inflammation on adult neurorespiratory control.

**eLife digest** Breathing is essential to life. At birth, the brain quickly adapts and learns to control breathing in different situations. This adaptability is called neuroplasticity. Most breathing-related adjustments in the brain are short-term, like breathing faster during exercise. The brain can also learn from prior experience to prepare for future situations. For example, intermittent exposure to low oxygen causes long-term changes in signals from the brain to muscles controlling breathing, which may help them prepare for future low oxygen situations. This is called long-term facilitation (LTF). This neuroplasticity may also help the brain to compensate or stabilize breathing during an illness or injury.

Illnesses shortly after birth can affect how the brain controls breathing and may contribute to respiratory diseases later in life. They may also have lasting effects on the ability to of the brain to learn and respond to stress, and may even contribute to psychiatric disorders or age-related cognitive decline.

Now, Hocker et al. show that inflammation shortly after birth has effects on breathing control that extend into adulthood. In the experiments, rats were injected four days after birth with either saline solution or a drug causing inflammation. When the rats grew into adults, their ability to make long-term breathing adjustments, or LTF, was assessed. In the rats exposed to early life inflammation two important pathways that enable LTF were eliminated. One pathway was restored when the rats received an anti-inflammatory treatment. Activating nerve cells reinstated the other pathway, suggesting these cells are not impaired.

The experiments suggest inflammation during early life impairs breathing control later on and may contribute to adult respiratory disease. Inflammation is common among infants in their first year, particularly among those born prematurely. This early-life inflammation may put them at risk of diseases associated with breathing control, like sleep apnoea, later in life. More studies are needed to understand the relationship between early life inflammation, respiratory control, and respiratory disease later in life.

DOI: https://doi.org/10.7554/eLife.45399.002

Respiratory plasticity is an important feature of the neural control of breathing, providing adaptability and maintenance of breathing when the respiratory system is challenged (*Fuller and Mitchell, 2017*). Phrenic long-term facilitation (pLTF) is a frequently studied adult model of respiratory motor plasticity (*Mitchell and Johnson, 2003*) and is elicited by at least two distinct cellular signaling pathways: the Q-pathway and the S-pathway (reviewed in *Dale-Nagle et al., 2010*). The Q-pathway is evoked by moderate acute intermittent hypoxia (mAIH; $3 \times 5$ min hypoxic episodes, $PaO_2$ 35–45 mmHg) and is serotonin dependent, while the S-pathway is evoked by severe AIH (sAIH, $PaO_2$ 25–35 mmHg) and is adenosine dependent (*Nichols et al., 2012*). Interestingly, Q-pathway-evoked plasticity is undermined by even low levels of acute, adult, systemic inflammation and restored by the non-steroidal anti-inflammatory, ketoprofen (*Vinit et al., 2011*; *Huxtable et al., 2013*; *Hocker and Huxtable, 2018*), while S-pathway-evoked adult plasticity is inflammation resistant (*Agosto-Marlin et al., 2017*). Though we are beginning to understand more about the mechanisms of acute, adult inflammation on respiratory motor plasticity (*Hocker et al., 2017*; *Hocker and Huxtable, 2018*), we do not know how inflammation in early postnatal life impacts respiratory motor plasticity in the adult. Furthermore, few studies have investigated sex-differences in pLTF (*Behan et al., 2002*; *Dougherty et al., 2017*) and we know even less about sex-differences in respiratory control in response to inflammation. Additionally, males are more sensitive acutely to neonatal inflammation leading to higher male mortality in neonates (*Bouman et al., 2005*; *Kentner et al., 2010*; *Rathod et al., 2017*), but our understanding of other sex-differences after neonatal inflammation are unknown. Given the profound effects of neonatal inflammation on other physiological systems, we tested the hypothesis that neonatal inflammation undermines Q-pathway, but not S-pathway, respiratory motor plasticity in adult male and female rats.

Our results indicate that one neonatal inflammatory challenge completely abolishes adult, AIH-induced Q-pathway and S-pathway respiratory motor plasticity. Despite no lasting increases in adult, inflammatory gene expression, Q-pathway impairment is inflammation-dependent and is restored by

acute adult anti-inflammatory treatment. Conversely, S-pathway impairment is inflammation-independent, but can be evoked by intermittent adenosine receptor agonism, suggesting phrenic motor neurons are not impaired. Since astrocytes are a primary source of adenosine during hypoxia (*Takahashi et al., 2010*; *Angelova et al., 2015*), they are likely impaired by neonatal inflammation and contributing to impairment of respiratory plasticity. These studies are the first steps toward understanding the lasting effects of neonatal inflammation on adult respiratory plasticity and suggest neonatal inflammation induces lasting-changes, increasing susceptibility to adult ventilatory control disorders.

## Results

### Neonatal inflammation acutely delays weight gain and increases male mortality

Male and female postnatal day 4 (P4) rats were injected with either LPS (Lipopolysaccharide; 1 mg/kg, i.p.) or saline. The dose of LPS was based on previous studies demonstrating CNS inflammatory gene expression in neonates (*Rourke et al., 2016*), as well as our unpublished data (N. Morrison, S. Johnson, J. Watters, A. Huxtable, unpublished observations). Within 24 hr of neonatal LPS injections, there was significantly greater mortality of male pups (8 of 67) than female pups (1 of 55, Fisher's exact test, p = 0.04, *Figure 1A*). No mortality was evident in the saline treated males (n = 63) or females (n = 63). For the surviving pups, neonatal LPS males weighed significantly less at week 7 (no pairwise weight differences seen in females), but importantly, weights were not different in adults (*Figure 1B*).

### Adult, Q-pathway-evoked pLTF was undermined by neonatal inflammation, and restored by acute, adult anti-inflammatory treatment

Q-pathway-evoked pLTF is evident as the increase in integrated phrenic activity 60 min after mAIH (PaO$_2$35–45 mmHg) in adult, anesthetized rats (*Bach and Mitchell, 1996*). As expected, in adult males treated with neonatal saline, Q-pathway-evoked pLTF was evident after mAIH (55 ± 33.2% change from baseline, n = 7, p = 0.0006, *Figure 2A and C*). However, Q-pathway-evoked pLTF was absent in adult males treated with neonatal LPS (14 ± 49%, n = 12, p = 0.2247 *Figure 2A and C*). To control for the known effects of estrus cycle hormones on pLTF in females (*Zabka et al., 2001*; *Behan et al., 2002*; *Dougherty et al., 2017*), adult females were ovariectomized 7–8 days before electrophysiology studies. Similar to males, adult females treated with neonatal saline displayed Q-pathway-evoked pLTF (97 ± 63% change from baseline, n = 7, p < 0.0001, *Figure 2B and C*), while adult females challenged with neonatal LPS did not express pLTF (−15 ± 43%, n = 6, p = 0.4689, *Figure 2B and C*). Phrenic amplitude did not change from baseline in the time control group (8 ± 6% change, n = 5, p = 0.6482), regardless of sex or neonatal LPS exposure, and was significantly reduced compared to males or females treated with neonatal saline. Between groups, Q-pathway-evoked pLTF was significantly abolished in adults after neonatal LPS compared to adults after neonatal saline for both males (p = 0.0200) males and females (p < 0.0001). Thus, neonatal inflammation induces lasting impairment of adult, Q-pathway-evoked respiratory motor plasticity in both males and females.

To test whether this lasting impairment of adult, respiratory motor plasticity was due to ongoing adult inflammation as a result of the neonatal inflammatory LPS challenge, we acutely treated adults with the non-steroidal anti-inflammatory, ketoprofen (12.5 mg/kg, i.p., 3 hr), a high dose previously shown to restore plasticity after acute, adult inflammation (*Huxtable et al., 2013*). Ketoprofen treatment restored Q-pathway-evoked pLTF in adult males treated with neonatal LPS (58 ± 18% change from baseline, n = 4, p = 0.0004, *Figure 3A and C*). Ketoprofen also restored Q-pathway-evoked pLTF in adult females treated with neonatal LPS (111 ± 44% from baseline, n = 5, p < 0.0001, *Figure 3B and C*). Adults treated with neonatal saline (male: 54 ± 17% from baseline, n = 4, 0.0008; female: 89 ± 40%, n = 5, p < 0.0001) were unaffected by adult ketoprofen treatment. Additionally, phrenic motor amplitude did not change in adult time controls treated with ketoprofen (13 ± 14% change from baseline, n = 4, p = 0.3436) and was significantly reduced compared to all other groups. Between groups, pLTF was not different between adult males (p = 0.7605) or females

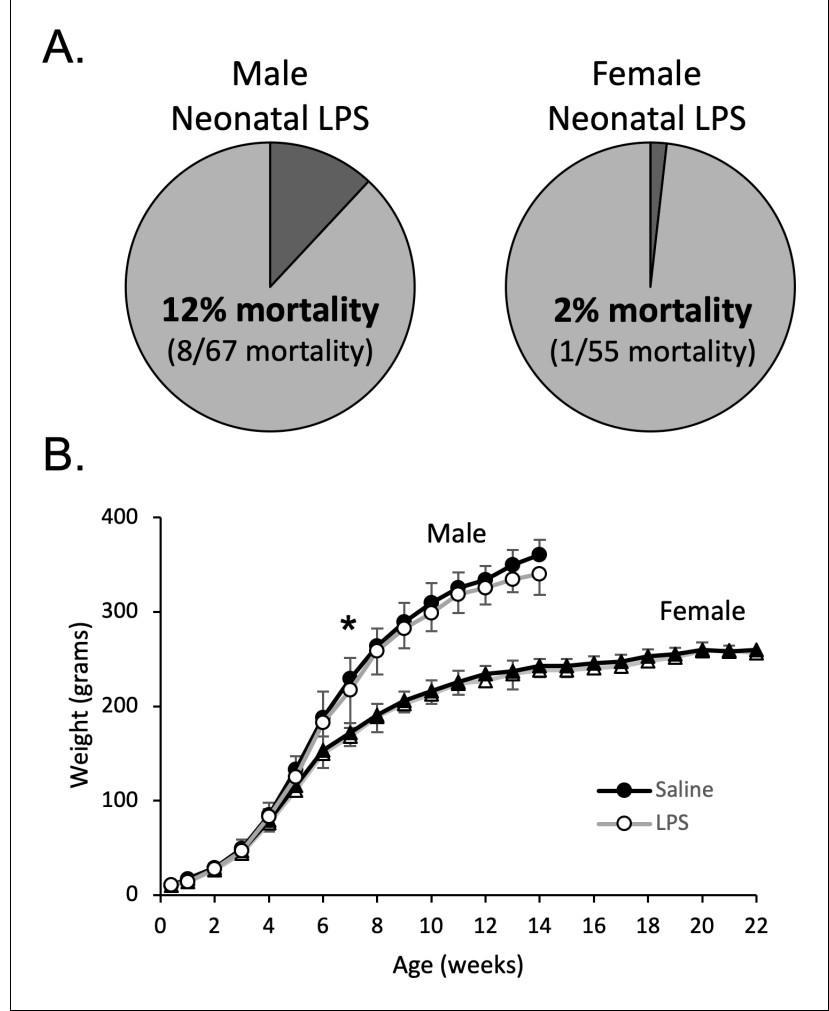

**Figure 1.** Neonatal inflammation increases mortality in neonatal males and transiently delays weight gain in male and female rats. After neonatal inflammation (P4, LPS 1 mg/kg, i.p.) male mortality (A) is increased within 24 hr (Fisher's exact test, p = 0.006), but not in females (p = 0.466), relative to saline controls. Weekly male and female weights (B) after neonatal saline or LPS. (*p < 0.05, significant pairwise difference within sex).
DOI: https://doi.org/10.7554/eLife.45399.003
The following source data is available for figure 1:

**Source data 1.** Weights.
DOI: https://doi.org/10.7554/eLife.45399.004

(p = 0. 2932) after neonatal saline or neonatal LPS, suggesting the impairment in Q-pathway-evoked pLTF is inflammation-dependent in both males and females.

## Neonatal inflammation did not induce chronic neuroinflammation in adult medulla or cervical spinal cords

Because the lasting impairment of Q-pathway-evoked pLTF was inflammation-dependent, we examined whether neonatal inflammation had lasting effects on adult neuroinflammation in regions involved in respiratory neural control and motor plasticity. Since plasticity was abolished in both males and females, data from both sexes were combined for analysis of inflammatory genes. In medullary and cervical spinal homogenates, neonatal LPS did not significantly alter mRNA for adult inflammatory genes (IL-6, IL-1β, TNF-α, or iNOS; *Figure 4A and B*). However, COX-2 gene expression was reduced in adult spinal cords after neonatal LPS (*Figure 4B*, p = 0.001), suggesting a decrease in COX-dependent inflammatory signaling. Thus, there was no evidence for lasting

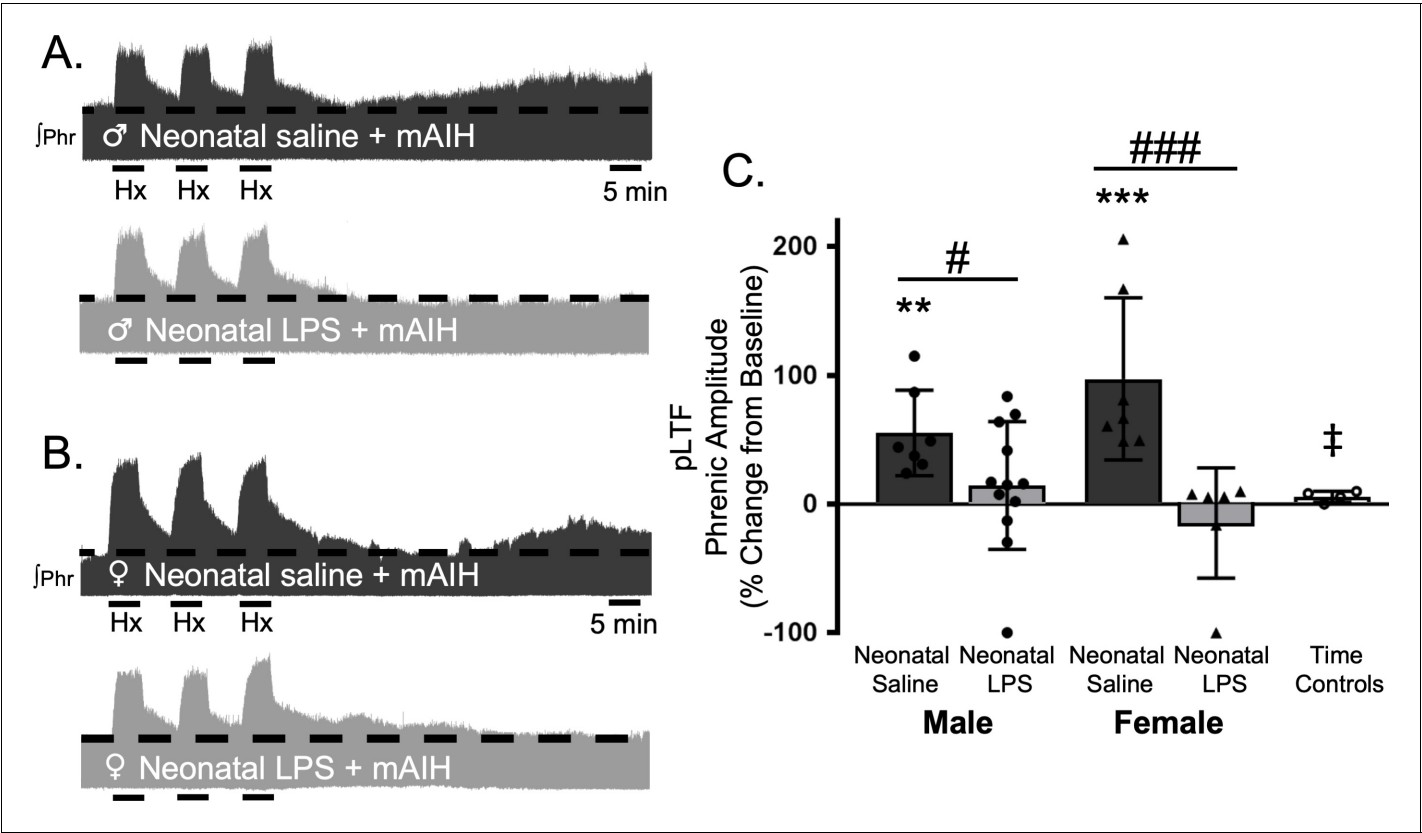

**Figure 2.** Neonatal systemic inflammation undermines adult, Q-pathway-evoked pLTF in male and female rats. Representative integrated phrenic neurograms from male (**A**) and female rats (**B**) after neonatal (P4) saline (top traces, black) or LPS (1 mg/kg, i.p.; bottom traces, grey). Q-pathway-evoked pLTF is evident in adults after neonatal saline as the progressive increase in phrenic nerve amplitude from baseline (dashed line) over 60 min following moderate acute intermittent hypoxia (mAIH, 3 × 5 min episodes, PaO$_2$35–45 mmHg). Group data (**C**) demonstrate Q-pathway-evoked pLTF 60 min after mAIH is abolished in adults by neonatal LPS in both males (circles) and females (triangles) and no change in phrenic amplitude in time controls (**p < 0.01, ***p < 0.001 from baseline, # p < 0.05, ### p < 0.001 between groups, ‡ p < 0.05 from adult males and females after neonatal saline).
DOI: https://doi.org/10.7554/eLife.45399.005
The following source data is available for figure 2:

**Source data 1.** mAIH.
DOI: https://doi.org/10.7554/eLife.45399.006

increases in neuroinflammatory gene expression in adults after a single exposure of neonatal inflammation in respiratory control regions.

### Adult S-pathway-evoked pLTF was undermined by neonatal inflammation, not restored by adult anti-inflammatory treatment, but revealed by intermittent adenosine 2A receptor agonism

S-pathway-evoked pLTF is evident as the increase in integrated phrenic activity 60 min after sAIH (PaO$_2$25–35 mmHg) in adult rats. As expected, in adult males after neonatal saline, S-pathway-evoked pLTF was evident after sAIH (61 ± 69% change from baseline, n = 5, p = 0.0001, *Figure 5A and C*). Contrary to our hypothesis, S-pathway-evoked pLTF was abolished in adult males after neonatal LPS (7 ± 18% change from baseline, n = 4, p = 0.6770, *Figure 5A and C*). In adult females after neonatal saline, S-pathway-evoked pLTF was evident (102 ± 47% change from baseline, n = 4, p < 0.0001, *Figure 5B and C*). Similar to adult males treated with neonatal LPS, S-pathway-evoked pLTF was abolished in adult females after neonatal LPS (0 ± 33%, n = 4, p = 0.9796, *Figure 5B and C*). Phrenic amplitude in the time control group was significantly less than males (p = 0.0147) or females (p < 0.0001) treated with neonatal saline. Between groups, S-pathway-evoked pLTF was significantly reduced after neonatal LPS in both adult males (p = 0.0180) and females (p < 0.0001)

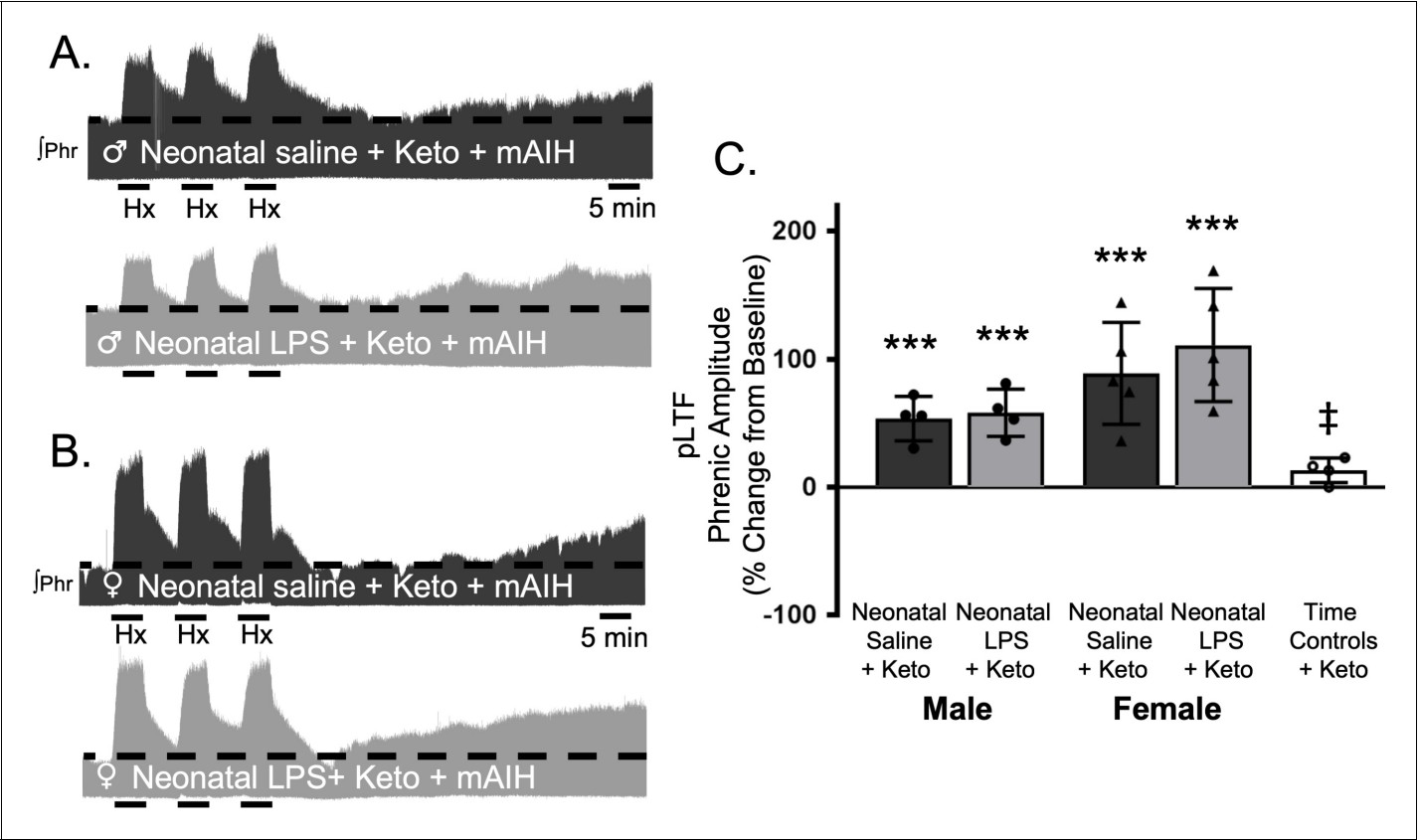

**Figure 3.** Acute, adult anti-inflammatory (ketoprofen, Keto) restores Q-pathway-evoked pLTF after neonatal systemic inflammation in adult male and female rats. Representative integrated phrenic neurograms for adult male (**A**) and female (**B**) rats after neonatal (P4) saline (top traces, black) or LPS (1 mg/kg, i.p.; bottom traces, grey) and acute, adult ketoprofen (12.5 mg/kg, i.p., 3 hr). Q-pathway-evoked pLTF is evident as the progressive increase in phrenic nerve amplitude from baseline (black dashed line) over 60 min following moderate acute intermittent hypoxia (mAIH, 3 × 5 min episodes, PaO$_2$35–45 mmHg). Group data (**C**) demonstrate adult ketoprofen restores Q-pathway-evoked pLTF 60 min after mAIH in adults after neonatal LPS in both males (circles) and females (triangles) and no change in phrenic amplitude in time controls (***p < 0.001 from baseline, ‡ p < 0.05 from all other groups).

DOI: https://doi.org/10.7554/eLife.45399.007

The following source data is available for figure 3:

**Source data 1.** mAIH_Keto.

DOI: https://doi.org/10.7554/eLife.45399.008

compared to adults after neonatal saline. Thus, neonatal inflammation induces lasting impairment of adult, S-pathway-evoked respiratory motor plasticity in both males and females.

To test whether this lasting impairment of adult, S-pathway-evoked plasticity is due to ongoing inflammation in adults after neonatal LPS, we examined sAIH-induced plasticity after an acute, adult treatment with ketoprofen (12.5 mg/kg, i.p., 3 hr). Ketoprofen did not alter normal expression of S-pathway-evoked pLTF in adult males after neonatal saline (63 ± 22% change from baseline, n = 5, p = 0.0014, *Figure 6A and C*). However, contrary to the Q-pathway results, adult ketoprofen did not restore S-pathway-evoked pLTF in adult males after neonatal LPS (0 ± 65% change from baseline, n = 5, p = 0.9804, *Figure 6A and C*). Similarly, adult females treated with neonatal saline also exhibited normal S-pathway-evoked pLTF after adult ketoprofen (130 ± 22% change from baseline, n = 5, 0.0803, *Figure 6B and C*), and S-pathway-evoked pLTF was not restored by ketoprofen in adult females after neonatal LPS (25 ± 30% change from baseline, n = 6, p = 0.0803, *Figure 6B and C*). Adult males and females treated with neonatal LPS and adult ketoprofen were not different from time controls (males, p = 0.4964; females p = 0.5227). Between groups, S-pathway-evoked pLTF after acute ketoprofen was significantly reduced in adults after neonatal LPS comapred to adults after neonatal saline in both males (p = 0.0019) and females (p < 0.0001). Thus, neonatal

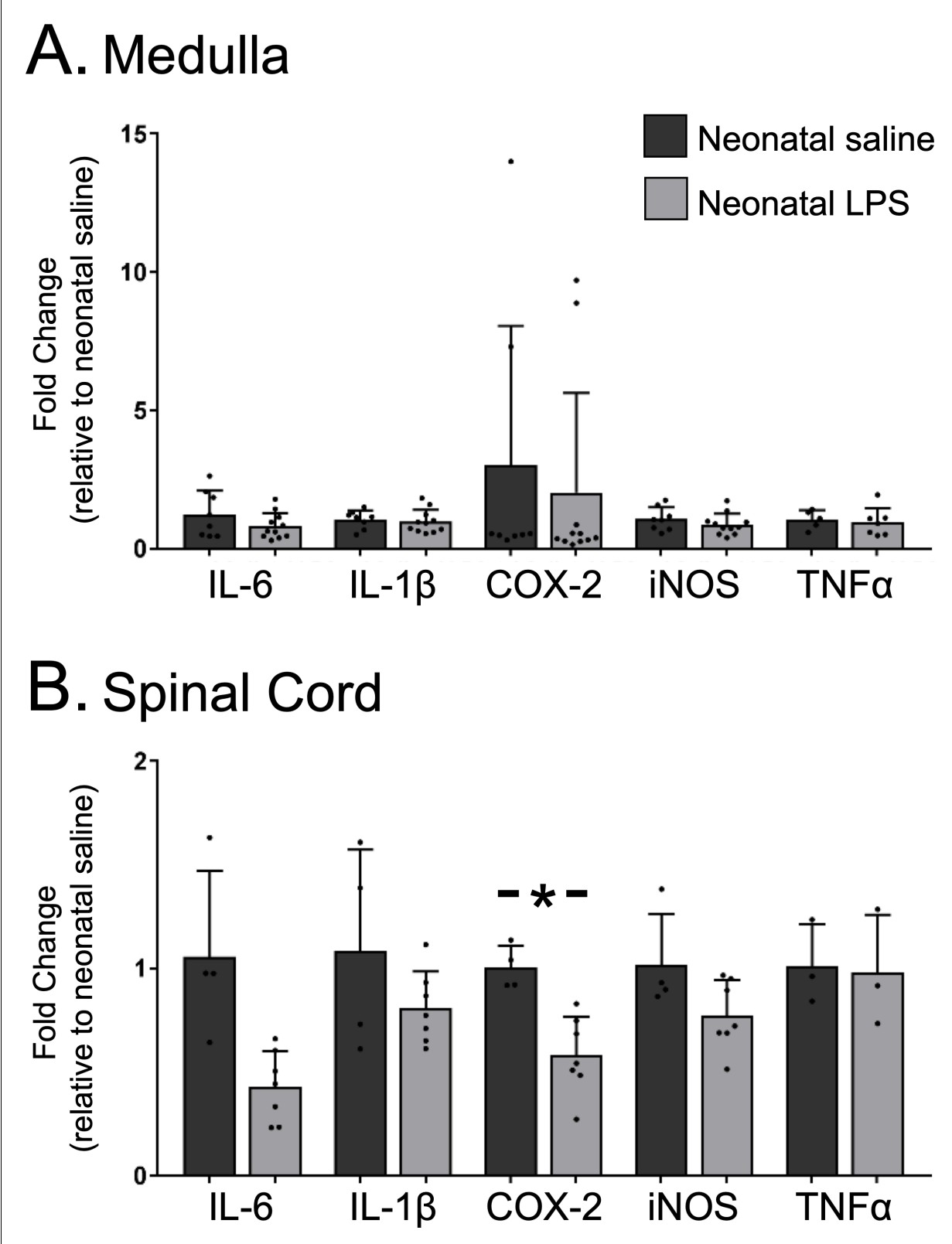

**Figure 4.** Neonatal inflammation does not increase adult medullary or spinal inflammatory gene expression. Homogenate samples isolated from adult medullas showed no significant increase in inflammatory mRNA after neonatal inflammation (**A**). Similarly, homogenate samples from isolated adult cervical spinal cords (**B**) were not increased by neonatal inflammation, but *COX2* gene expression was significantly decreased in adults after neonatal inflammation (*p < 0.05).

*Figure 4 continued on next page*

*Figure 4 continued*

DOI: https://doi.org/10.7554/eLife.45399.009

The following source data is available for figure 4:

**Source data 1.** Cytokine_expression.

DOI: https://doi.org/10.7554/eLife.45399.010

inflammation induces a lasting impairment of adult, S-pathway-evoked respiratory motor plasticity, which is not due to ongoing adult, inflammatory signaling.

S-pathway-evoked plasticity elicited by sAIH is adenosine dependent (*Golder et al., 2008*; *Nichols et al., 2012*) and can be evoked by intermittent CGS-21680, an adenosine 2A receptor agonist. To test if neonatal inflammation is impairing phrenic motor neurons and preventing pLTF, we examined phrenic output after intermittent CGS-21680 on the cervical spinal cord, around the phrenic motor pool. Intrathecal CGS-21680 (100 µM, 3 × 10 µL) evoked phrenic motor plasticity in adult males after neonatal saline (110 ± 17% change from baseline, n = 4, p < 0.001, *Figure 7A and*

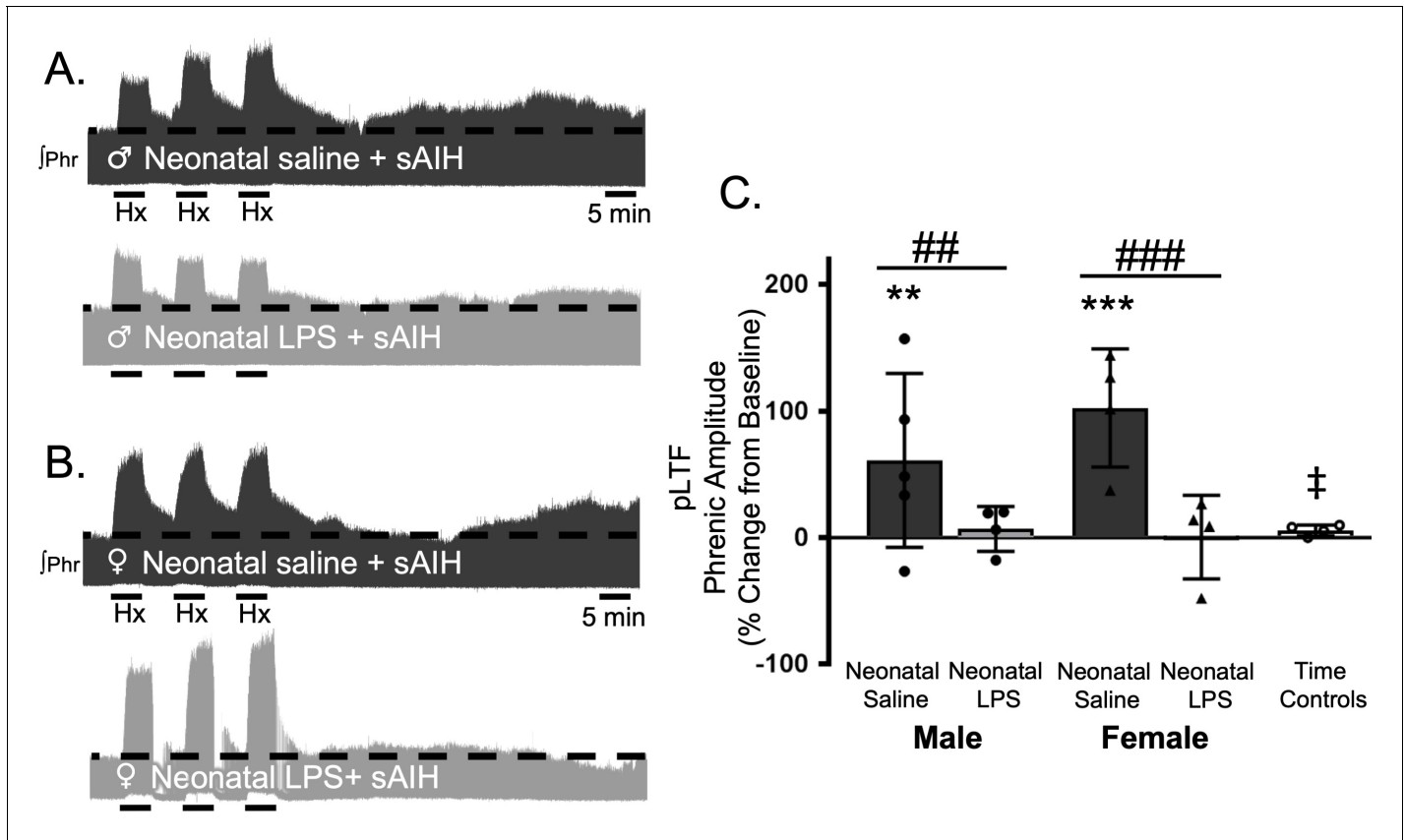

**Figure 5.** Neonatal systemic inflammation undermines adult, S-pathway-evoked pLTF in male and female rats. Representative integrated phrenic neurograms for adult male (A) and female (B) rats after neonatal (P4) saline (top traces, black) or LPS (1 mg/kg, i.p.; bottom traces, grey). S-pathway-evoked pLTF is evident as the progressive increase in phrenic nerve amplitude from baseline (black dashed line) over 60 min following severe acute intermittent hypoxia (sAIH, 3 × 5 min episodes, PaO$_2$ 25–35 mmHg) in adults after neonatal saline. Group data (C) demonstrate S-pathway-evoked pLTF 60 min after sAIH is abolished in adults by neonatal LPS in both males (circles) and females (triangles) and no change in phrenic amplitude in time controls (**p < 0.01, ***p < 0.001 from baseline ## p < 0.01, ### p < 0.001 between groups, ‡ p < 0.05 from male and female adults after neonatal saline).

DOI: https://doi.org/10.7554/eLife.45399.011

The following source data is available for figure 5:

**Source data 1.** sAIH.

DOI: https://doi.org/10.7554/eLife.45399.012

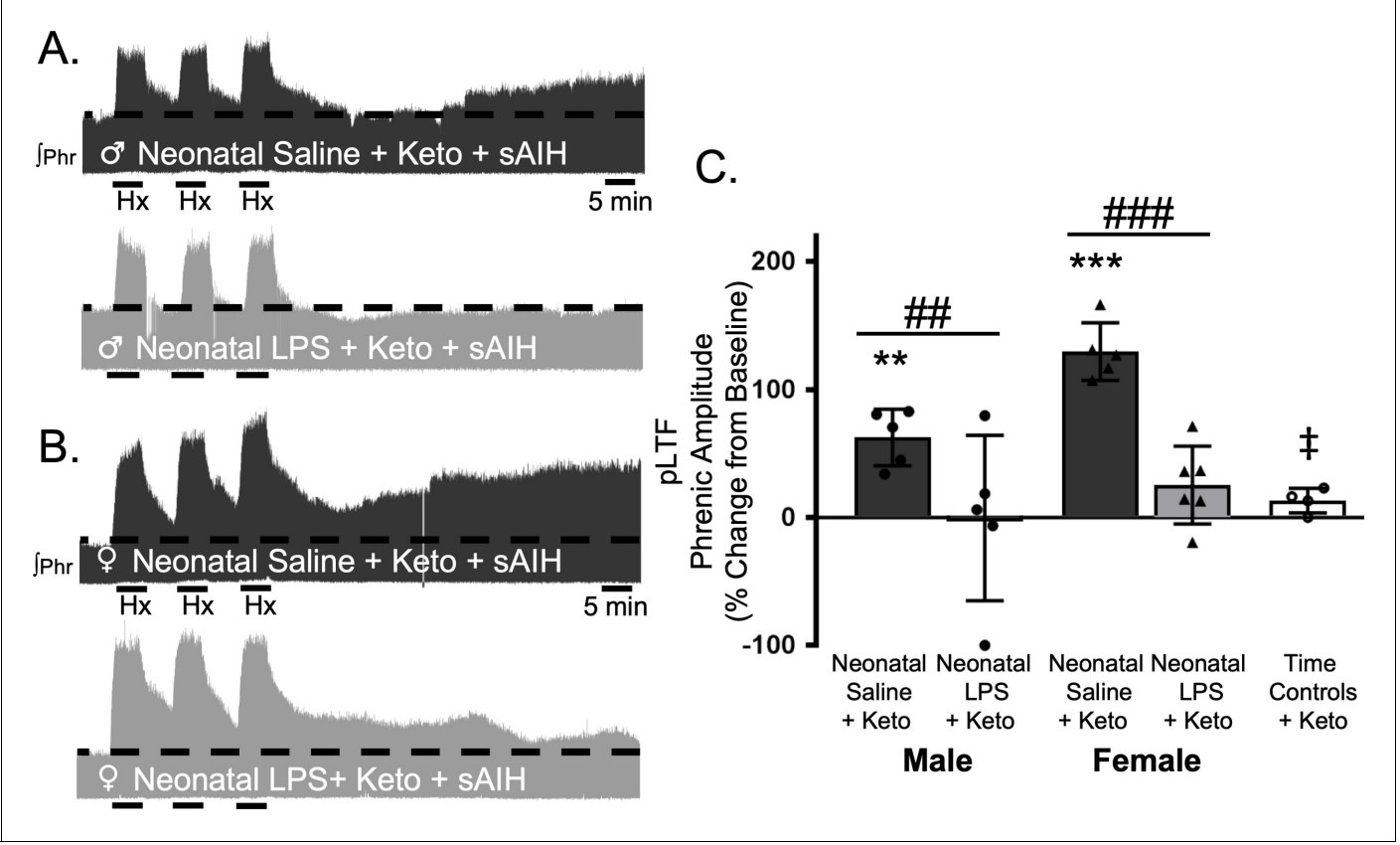

**Figure 6.** Adult, anti-inflammatory (ketoprofen, keto) does not restore S-pathway-evoked pLTF after neonatal systemic inflammation in adult male and female rats. Representative integrated phrenic neurograms for adult male (**A**) and female (**B**) rats after neonatal (P4) saline (top traces, black) or LPS (1 mg/kg, i.p.; bottom traces, grey) and acute, adult ketoprofen (12.5 mg/kg, i.p., 3 hr). S-pathway-evoked pLTF is evident as the progressive increase in phrenic nerve amplitude from baseline (black dashed line) over 60 min following severe acute intermittent hypoxia (sAIH, 3 × 5 min episodes, PaO$_2$35–45 mmHg) in adults after neonatal saline. Group data (**C**) demonstrate acute, adult ketoprofen does not restore S-pathway-evoked pLTF 60 min after sAIH after neonatal LPS in adult males (circles) and females (triangles) and no change in phrenic amplitude in time controls (**p < 0.01, ***p < 0.001 from baseline, ## p < 0.01, ### p < 0.001 between groups, ‡ p < 0.05 from adult males and females after neonatal saline).
DOI: https://doi.org/10.7554/eLife.45399.013

The following source data is available for figure 6:

**Source data 1.** sAIH_Keto.
DOI: https://doi.org/10.7554/eLife.45399.014

**C**) and females after neonatal saline (127 ± 47%, n = 4, p < 0.001, *Figure 7B and C*). After neonatal LPS, intrathecal CGS-21680 also elicited plasticity in adult males (85 ± 64%, n = 6, p < 0.001, *Figure 7A and C*) and adult females (147 ± 74%, n = 6, p < 0.001, *Figure 7B and C*), demonstrating phrenic motor neurons are not impaired after neonatal inflammation and are capable of S-pathway-evoked plasticity. The vehicle control group was not different from baseline (−3 ± 5% change, n = 4, p = 0.8891) and significantly reduced compared to all other groups. Between groups, pLTF was not different between adult males (p = 0.2841) or females (p = 0.4032) after neonatal saline or neonatal LPS. Thus, adult phrenic motor neurons are not impaired after neonatal inflammation and are capable of plasticity after neonatal inflammation. Therefore, the source of intermittent adenosine release is impaired during sAIH-induced pLTF after neonatal inflammation.

## Adult microglia and astrocyte density were not changed by neonatal inflammation

While there was no evidence for elevated neuroinflammation based on the inflammatory genes evaluated here, the anti-inflammatory drug ketoprofen successfully restored Q-pathway-evoked plasticity. Additionally, our results indicate the impairment in S-pathway-evoked plasticity was likely due to

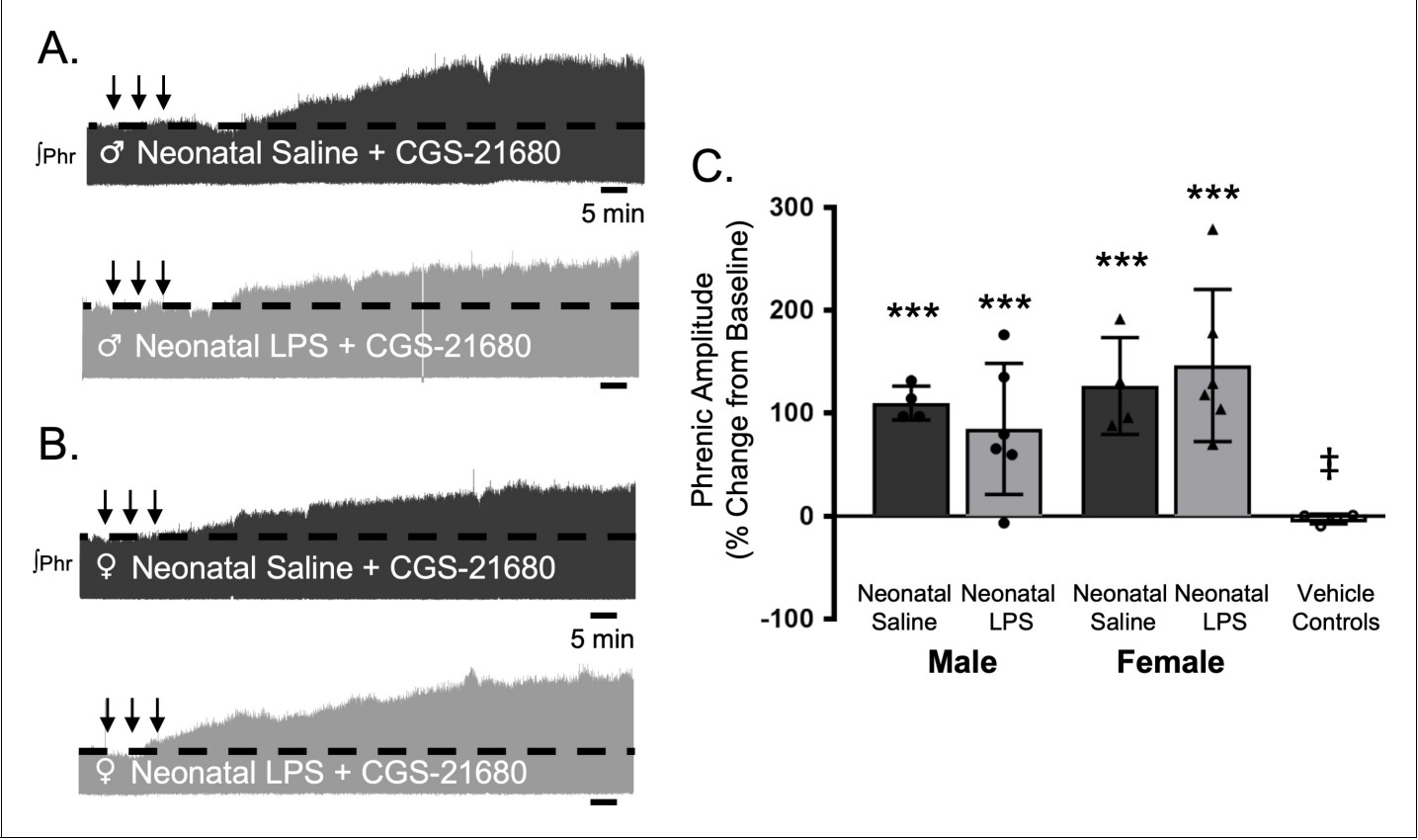

**Figure 7.** Intermittent adult, adenosine receptor agonism reveals plasticity after neonatal systemic inflammation in male and female rats. Representative integrated phrenic neurograms for adult male (**A**) and female (**B**) rats after neonatal (P4) saline (top traces, black) or LPS (1 mg/kg, i.p.; bottom traces, grey). S-pathway-evoked phrenic motor plasticity is evident as the progressive increase in phrenic nerve amplitude from baseline (black dashed line) 90 min following intermittent CGS-21680 (100 μM, black arrows, 3 × 5 min apart) in adults after neonatal saline. Group data (**C**) demonstrate adult CGS-21680 reveals S-pathway-evoked plasticity after neonatal LPS in adult males (circles) and females (triangles) and no change in phrenic amplitude in vehicle controls (***p < 0.001 from baseline, ‡ p < 0.001 from adult males and females after neonatal saline).
DOI: https://doi.org/10.7554/eLife.45399.015

The following source data is available for figure 7:

**Source data 1.** CGS2160.
DOI: https://doi.org/10.7554/eLife.45399.016

a lasting change in adenosine signaling, possibly as a result of altered astrocytes. Thus, we hypothesized a lasting change in astrocytes and microglia in respiratory control regions, influencing neuronal function and impairing adult plasticity. We evaluated GFAP (astrocytes) and IBA1 (microglia) immunoreactivity in the adult preBötC, the site of respiratory rhythmogenesis (*Smith et al., 1991*), and in cervical spinal cords in the region of the phrenic motor nucleus, the presumptive site of pLTF (*Baker-Herman et al., 2004*; *Devinney et al., 2015*; *Dale et al., 2017*). Neonatal inflammation did not alter GFAP (p = 0.5969) or IBA1 (p = 0.6487) immunoreactivity in adult preBötC in either sex (*Figure 8A,B and E*), suggesting astrocyte and microglial density were not changed in adults after neonatal inflammation. Furthermore, there were no changes in GFAP (p = 0.7195) or IBA1 (p = 0.9254) immunoreactivity in adult cervical spinal cords (*Figure 8C,D and F*), suggesting no lasting changes in astrocyte and microglia density in the region of the phrenic motor nucleus. Additionally, no obvious differences in astrocyte or microglial morphology in adult phrenic motor nuclei or the preBötC were seen following neonatal LPS inflammation, suggesting other signaling mechanisms are responsible for impairing adult pLTF.

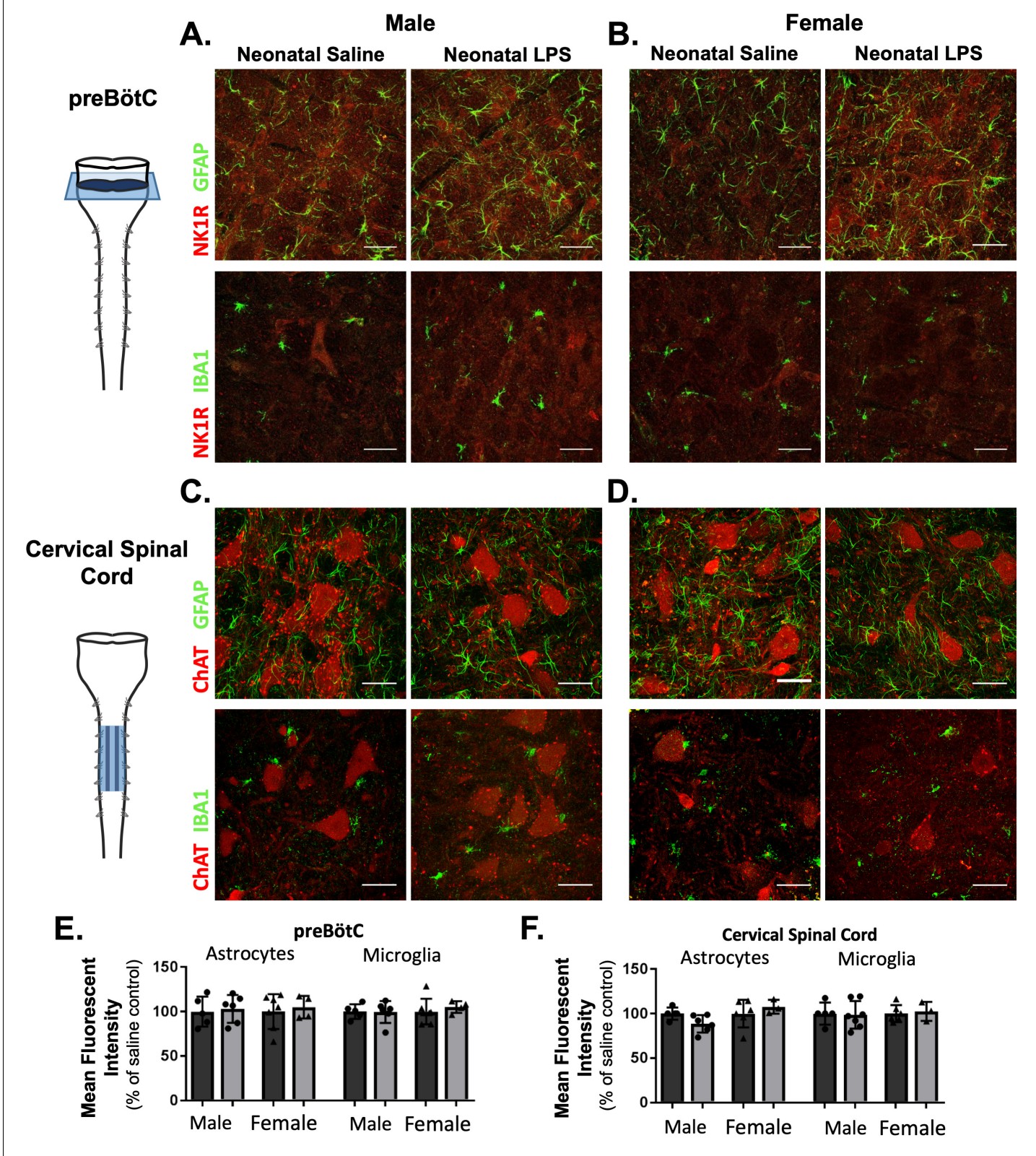

**Figure 8.** Neonatal inflammation does not alter GFAP or IBA1 immunofluorescence in adult preBötzinger Complex or ventral cervical spinal cords. After neonatal LPS (1 mg/kg, i.p., (P4), representative confocal images (40x) from adult preBötC (**A and B**) and cervical spinal cords (**C and D**) displayed no qualitative differences in immunoreactivity for GFAP (green, astrocytes) or IBA1 (green, microglia) in males (left panels) or females (right panels). PreBötC neurons are labeled with antibodies for NK1R (red, **A and B**) and motor neurons are labeled with antibodies for ChAT (red, **C and D**). Neonatal

*Figure 8 continued on next page*

*Figure 8 continued*
inflammation did not significantly change mean fluorescent intensity of either GFAP or IBA1 in the preBötC (**E**) or cervical spinal cord (**F**), suggesting no lasting differences in astrocytes or microglia after neonatal inflammation. Scale bars: 50 µm.

DOI: https://doi.org/10.7554/eLife.45399.017

The following source data is available for figure 8:

**Source data 1.** Immunohistochemistry.

DOI: https://doi.org/10.7554/eLife.45399.018

## Acute hypoxic phrenic responses were greater in females, but were unaffected by neonatal inflammation

Neonatal inflammation did not significantly alter moderate acute hypoxic phrenic amplitude responses within adult males (neonatal saline = 114 ± 41% change from baseline; neonatal LPS = 93 ± 36%) or females (neonatal saline = 185 ± 53%; neonatal LPS = 148 ± 63%, *Table 1*). Hypoxic phrenic amplitude responses were also unaffected by the anti-inflammatory ketoprofen in adult males (neonatal saline +Keto = 118 ± 36%; neonatal LPS +Keto, 118 ± 44%) or females (neonatal saline +Keto, 165 ± 52%; neonatal LPS +Keto, 189 ± 82%, *Table 1*). However, adult females exhibited significantly greater acute phrenic amplitude responses to moderate hypoxia (main effect, p = 0.0004).

Phrenic amplitude in response to severe hypoxia was similarly unaltered by neonatal inflammation within adult males (neonatal saline = 139 ± 37% change from baseline; neonatal LPS = 106 ± 10%) or females (neonatal saline = 172 ± 125%; neonatal LPS = 172 ± 26%, *Table 1*). Acute ketoprofen pretreatment did not alter acute hypoxic phrenic amplitude responses in adult males (neonatal saline = 151 ± 25% change from baseline; neonatal LPS, 174 ± 96%) or females (neonatal saline = 194 ± 45%; neonatal LPS = 235 ± 63%, *Table 1*). Similarly, adult females exhibited a significantly greater acute amplitude response to severe hypoxia than males (main effect, p = 0.021).

**Table 1.** Acute, adult hypoxic phrenic responses.

| | Male | | Female ††† | |
|---|---|---|---|---|
| | **Neonatal saline** | **Neonatal LPS** | **Neonatal saline** | **Neonatal LPS** |
| Moderate hypoxia | 114 ± 41 | 93 ± 36 | 185 ± 53[*] | 148 ± 63 |
| Keto + Moderate hypoxia | 118 ± 36 | 118 ± 44 | 165 ± 52 | 189 ± 82[*] |
| | **Male** | | **Female †** | |
| | **Neonatal Saline** | **Neonatal LPS** | **Neonatal Saline** | **Neonatal LPS** |
| Severe hypoxia | 139 ± 37 | 106 ± 10 | 172 ± 125 | 172 ± 26 |
| Keto + Severe hypoxia | 151 ± 25 | 174 ± 96 | 194 ± 45 | 235 ± 63 |

Group data for adult, acute hypoxic phrenic responses to moderate (PaO$_2$35–45 mmHg) and severe (PaO$_2$25–35 mmHg) hypoxia demonstrate no differences after neonatal (P4) saline or LPS (1 mg/kg, i.p), or after adult ketoprofen (12.5 mg/kg, i.p, 3 hr) within each sex. Significant differences between sexes demonstrate larger responses in females after moderate or severe hypoxia († p<0.05, ††† p<0.001. *p<0.05 from male neonatal LPS. Moderate hypoxia: neonatal saline male (n = 7), neonatal LPS male (n = 10), neonatal saline female (n = 6), neonatal LPS female (n = 6). Keto + Moderate hypoxia: neonatal saline male (n = 4), neonatal LPS male (n = 4), neonatal saline female (n = 5), neonatal LPS female (n = 5). Severe hypoxia: neonatal saline male (n = 5), neonatal LPS male (n = 4), neonatal saline female (n = 4), neonatal LPS female (n = 4). Keto + Moderate hypoxia: neonatal saline male (n = 5), neonatal LPS male (n = 5), neonatal saline female (n = 5), neonatal LPS female (n = 6)

DOI: https://doi.org/10.7554/eLife.45399.019

The following source data is available for Table 1:

**Source data 1.** Hypoxic responses.

DOI: https://doi.org/10.7554/eLife.45399.020

## Physiological parameters and frequency plasticity

All physiological parameters remained within experimental limits (*Table 2*). Neonatal saline or LPS caused no significant changes in adult temperature, $PaCO_2$, $PaO_2$, or pH at baseline. There were no between group differences in baseline MAP, suggesting no long-lasting cardiovascular changes after neonatal inflammation. No significant changes occurred over time in temperature, pH, or $PaCO_2$ for any group. As expected, MAP and $PaO_2$ were significantly decreased during hypoxic episodes in experimental groups (*Huxtable et al., 2015*; *Hocker and Huxtable, 2018*), but these changes were not evident in time control groups and were not different from baseline values at 60 min post-AIH.

Baseline phrenic burst frequency was not significantly different between groups and frequency plasticity, an increase in burst frequency 60 min after AIH (*Baker-Herman and Mitchell, 2008*), was not evident in any group. Phrenic burst frequency did not change after intrathecal CGS-21680.

## Discussion

Although neonatal inflammation is common (*Stoll et al., 2002*; *Stoll et al., 2004*), little is known concerning how neonatal inflammation alters ventilatory control. Here, we investigated the long-term consequences of neonatal systemic inflammation on adult respiratory motor plasticity, a key feature of the neural control of breathing providing adaptability to respiratory system challenges (*Mitchell and Johnson, 2003*). We show for the first time that a single inflammatory challenge to neonates completely abolishes AIH-induced Q-pathway and S-pathway-evoked respiratory motor plasticity in adult males and females. Our results indicate a persistent change in adult inflammatory signaling contributes to this impairment since adult anti-inflammatory treatment restores Q-pathway-evoked, but not S-pathway-evoked, pLTF. Further, this is the first evidence of impairment of S-pathway-evoked motor plasticity, suggesting neonatal inflammation likely leads to a further vulnerable adult as this pathway was thought of as a 'backup pathway' after acute, adult inflammation (*Agosto-Marlin et al., 2017*). However, we demonstrate S-pathway plasticity can be revealed by intermittent, spinal adenosine receptor agonism, suggesting astrocyte dysfunction after neonatal inflammation since they are the likely source of adenosine during hypoxia (*Takahashi et al., 2010*; *Angelova et al., 2015*). These studies are the first steps toward understanding the lasting effects of neonatal inflammation on adult neurorespiratory control and suggest neonatal inflammation may increase susceptibility to adult ventilatory control disorders.

LPS-induced neonatal inflammation transiently upregulates cytokines in regions involved in respiratory control and plasticity (N. Morrison, S. Johnson, J. Watters, A. Huxtable, unpublished observations), consistent with inflammatory profiles in other CNS regions (*Wang et al., 2006*; *Schwarz and Bilbo, 2011*; *Bilbo and Schwarz, 2012*; *Jafri et al., 2013*). Neonatal inflammation also increases male mortality, consistent with clinical male mortality after neonatal inflammation (*Person et al., 2014*) and is relevant to the increased risk of sudden infant death syndrome for males (*Kinney and Thach, 2009*). However, similar to other studies (*Bilbo et al., 2005b*; *Mouihate et al., 2010*; *Smith et al., 2014*), we found no measurable adult changes in cytokines, or glial number or morphology despite a lasting inflammation-dependent impairment in Q-pathway-evoked pLTF. While we have previously demonstrated adult Q-pathway pLTF is sensitive to low levels of acute, systemic inflammation (*Huxtable et al., 2011*; *Huxtable et al., 2013*; *Huxtable et al., 2015*; *Huxtable et al., 2018a*; *Vinit et al., 2011*; *Hocker and Huxtable, 2018*), this is the first study to demonstrate neonatal inflammation induces lasting changes in inflammatory signaling to undermine adult Q-pathway-evoked pLTF. Additionally, S-pathway-evoked pLTF is not restored by even the high dose of ketoprofen used here, but is revealed by intermittent adenosine 2A receptor agonism on phrenic motor neurons (*Seven et al., 2018*). Thus, these results indicate phrenic motor neurons are not impaired after neonatal inflammation and the loss of S-pathway-evoked plasticity is likely due to impaired adenosine signaling during hypoxia. Furthermore, a likely source of adenosine during hypoxia is astrocytes (*Takahashi et al., 2010*; *Angelova et al., 2015*), suggesting neonatal inflammation induces lasting astrocyte-specific changes in the adult spinal cord to impair adult S-pathway-evoked pLTF. Further understanding the mechanisms impairing distinct forms of respiratory motor plasticity is required to develop plasticity as a therapeutic tool, such as for spinal cord injury and amyotrophic lateral sclerosis (*Mitchell, 2008*; *Gonzalez-Rothi et al., 2015*). Additionally, considering the crosstalk between Q- and S-pathways (*Dale-Nagle et al., 2010*; *Fields and Mitchell, 2015*; *Perim et al., 2018*), the response of the respiratory control system likely depends on the functional status of both

**Table 2.** Physiological parameters during electrophysiology experiments.

| | Temperature (°C) | | $P_aO_2$ (mmHg) | | $P_aCO_2$ (mmHg) | | pH | | MAP (mmHg) | |
|---|---|---|---|---|---|---|---|---|---|---|
| **Baseline** | **Male** | **Female** | **Male** | **Female** | **Male** | **Female** | **Male** | **Female** | **Male** | **Female** |
| Neonatal Saline + mAIH | 37.4 ± 0.2 | 37.4 ± 0.2 | 254 ± 19 | 259 ± 53 | 43.2 ± 5.6 | 48.9 ± 3.7 | 7.37 ± 0.06 | 7.36 ± 0.02 | 124 ± 9 | 121 ± 18 |
| Neonatal LPS + mAIH | 37.6 ± 0.2 | 37.5 ± 0.2 | 266 ± 30 | 268 ± 24 | 42.8 ± 4.7[#] | 45.0 ± 1.8 | 7.37 ± 0.04 | 7.37 ± 0.02 | 127 ± 10 | 123 ± 23 |
| Neonatal Saline + Keto + mAIH | 37.3 ± 0.1 | 37.4 ± 0.3 | 249 ± 21 | 255 ± 28 | 41.7 ± 4.9 | 48.6 ± 3.5 | 7.38 ± 0.03 | 7.33 ± 0.02 | 132 ± 8 | 121 ± 12 |
| Neonatal LPS + Keto + mAIH | 37.4 ± 0.2 | 37.4 ± 0.3 | 276 ± 40 | 283 ± 33 | 41.5 ± 3.2 | 47.9 ± 3.6 | 7.39 ± 0.03 | 7.34 ± 0.02 | 129 ± 16 | 117 ± 14 |
| Neonatal Saline + sAIH | 37.6 ± 0.3 | 37.5 ± 0.2 | 295 ± 18 | 266 ± 9 | 43.3 ± 5.9 | 47.7 ± 3.1 | 7.37 ± 0.02 | 7.36 ± 0.00 | 133 ± 6 | 121 ± 19 |
| Neonatal LPS + sAIH | 37.4 ± 0.1 | 37.4 ± 0.4 | 297 ± 28 | 264 ± 35 | 45.3 ± 4.2 | 47.7 ± 4.0 | 7.36 ± 0.02 | 7.36 ± 0.02 | 135 ± 20 | 132 ± 14 |
| Neonatal Saline + Keto + sAIH | 37.4 ± 0.3 | 37.5 ± 0.1 | 256 ± 46 | 268 ± 29 | 41.6 ± 2.2[e] | 49.9 ± 3.5 | 7.39 ± 0.03 | 7.34 ± 0.02 | 110 ± 12 | 113 ± 29 |
| Neonatal LPS + Keto + sAIH | 37.5 ± 0.2 | 37.4 ± 0.2 | 243 ± 45 | 245 ± 39 | 42.3 ± 3.4 | 51.6 ± 6.8 | 7.37 ± 0.01 | 7.31 ± 0.04 | 121 ± 3 | 129 ± 13 |
| Neonatal Saline + CGS-21680 | 37.5 ± 0.2 | 37.3 ± 0.2 | 268 ± 21 | 237 ± 14 | 45.4 ± 5.3 | 46.2 ± 3 | 7.36 ± 0.04 | 7.34 ± 0.03 | 119 ± 24 | 121 ± 10 |
| Neonatal LPS + CGS-21680 | 37.6 ± 0.3 | 37.4 ± 0.3 | 247 ± 27 | 247 ± 21 | 46.2 ± 2.9 | 46.6 ± 2.7 | 7.37 ± 0.02 | 7.34 ± 0.02 | 114 ± 12 | 107 ± 15 |
| Time Controls | 37.7 ± 0.1 | | 250 ± 45 | | 49.4 ± 4.0 | | 7.36 ± 0.05 | | 110 ± 10 | |
| Time Controls + Keto | 37.5 ± 0.3 | | 250 ± 17 | | 44.0 ± 9.1 | | 7.37 ± 0.06 | | 115 ± 39 | |
| CGS-21680 Vehicle Controls | 37.4 ± 0.3 | | 237 ± 40 | | 45.8 ± 0.8 | | 7.37 ± 0.03 | | 110 ± 17 | |
| **Hypoxia** | | | | | | | | | | |
| Neonatal Saline + mAIH | 37.4 ± 0.2 | 37.3 ± 0.1 | 38 ± 2[*,†,‡] | 40 ± 3[*,†,‡] | 42.2 ± 5.1[#] | 48.8 ± 3.8 | 7.35 ± 0.07 | 7.36 ± 0.03[#] | 62 ± 23[*,†,‡] | 65 ± 30[*,†,‡] |
| Neonatal LPS + mAIH | 37.5 ± 0.4 | 37.4 ± 0.1 | 39 ± 2[*,†,‡] | 39 ± 4[*,†,‡] | 43.0 ± 4.8[#] | 45.4 ± 2.8 | 7.36 ± 0.04[#] | 7.36 ± 0.02[#] | 68 ± 19[*,†,‡] | 80 ± 21[‡] |
| Neonatal Saline + Keto + mAIH | 37.5 ± 0.2 | 37.4 ± 0.3 | 38 ± 1[*,†,‡] | 39 ± 3[*,†,‡] | 41.9 ± 3.5 | 48.6 ± 3.7 | 7.37 ± 0.03[#] | 7.32 ± 0.04 | 62 ± 11[‡] | 56 ± 6[*,†,‡] |
| Neonatal LPS + Keto + mAIH | 37.5 ± 0.2 | 37.5 ± 0.3 | 40 ± 2[*,†,‡] | 39 ± 4[*,†,‡] | 40.9 ± 2.2[#] | 47.8 ± 4.9 | 7.36 ± 0.04 | 7.32 ± 0.03 | 71 ± 14[‡] | 66 ± 19[‡] |
| Neonatal Saline + sAIH | 37.6 ± 0.2 | 37.4 ± 0.3 | 29 ± 5[*,†,‡] | 29 ± 2[*,†,‡] | 43.5 ± 5.9 | 47.7 ± 2.3 | 7.35 ± 0.03 | 7.32 ± 0.06 | 58 ± 9[*,†,‡] | 53 ± 11[*,†,‡] |
| Neonatal LPS + sAIH | 37.4 ± 0.3 | 37.5 ± 0.3 | 30 ± 4[*,†,‡] | 31 ± 5[*,†,‡] | 46.3 ± 4.2 | 47.3 ± 5.9 | 7.34 ± 0.03 | 7.31 ± 0.03 | 61 ± 20[‡] | 59 ± 20[*,†,‡] |
| Neonatal Saline + Keto + sAIH | 37.4 ± 0.2 | 37.5 ± 0.2 | 30 ± 2[*,†,‡] | 32 ± 3[*,†,‡] | 42.3 ± 2.2[#] | 48.9 ± 3.7 | 7.36 ± 0.03 | 7.29 ± 0.03 | 34 ± 8[*,†,‡,¶] | 45 ± 29[*,†,‡] |
| Neonatal LPS + Keto + sAIH | 37.3 ± 0.3 | 37.6 ± 0.2 | 31 ± 2[*,†,‡] | 32 ± 1[*,†,‡] | 42.2 ± 3.2[#] | 52.2 ± 5.8 | 7.34 ± 0.03 | 7.28 ± 0.06 | 37 ± 11[*,†,‡,¶] | 43 ± 21[*,†,‡] |
| Time Controls | 37.6 ± 0.3 | | 226 ± 40 | | 48.7 ± 4.7 | | 7.35 ± 0.04 | | 107 ± 13 | |
| Time Controls + Keto | 37.5 ± 0.2 | | 258 ± 13 | | 45.5 ± 9.3 | | 7.37 ± 0.06[#] | | 109 ± 44 | |
| **60 min** | | | | | | | | | | |
| Neonatal Saline + mAIH | 37.5 ± 0.4 | 37.3 ± 0.1 | 234 ± 28 | 259 ± 22 | 43.4 ± 5.7 | 48.6 ± 3.7 | 7.38 ± 0.05[#] | 7.35 ± 0.02 | 114 ± 9 | 117 ± 27 |
| Neonatal LPS + mAIH | 37.5 ± 0.3 | 37.4 ± 0.3 | 253 ± 19 | 268 ± 23[*] | 42.9 ± 4.4[#] | 45.2 ± 2.6 | 7.39 ± 0.04[#] | 7.37 ± 0.01 | 116 ± 14 | 121 ± 25 |
| Neonatal Saline + Keto + mAIH | 37.3 ± 0.2 | 37.3 ± 0.3 | 262 ± 14 | 257 ± 32 | 42.5 ± 4.9 | 48.7 ± 4.0 | 7.38 ± 0.01 | 7.33 ± 0.04 | 121 ± 13 | 115 ± 11 |
| Neonatal LPS + Keto + mAIH | 37.6 ± 0.3 | 37.6 ± 0.3 | 257 ± 18 | 276 ± 40[*] | 41.5 ± 2.7 | 47.8 ± 3.6 | 7.36 ± 0.02 | 7.34 ± 0.06 | 123 ± 11 | 112 ± 18 |
| Neonatal Saline + sAIH | 37.5 ± 0.3 | 37.4 ± 0.2 | 262 ± 36 | 258 ± 19 | 43.7 ± 5.5 | 47.6 ± 2.9 | 7.37 ± 0.04 | 7.32 ± 0.03 | 135 ± 9 | 115 ± 24 |
| Neonatal LPS + sAIH | 37.7 ± 0.2 | 37.4 ± 0.2 | 282 ± 16[*] | 266 ± 18 | 46.2 ± 4.8 | 47.7 ± 4.5 | 7.36 ± 0.02 | 7.35 ± 0.02 | 127 ± 14 | 128 ± 17 |
| Neonatal Saline + Keto + sAIH | 37.7 ± 0.3 | 37.4 ± 0.3 | 248 ± 42 | 262 ± 9 | 42.3 ± 2.6 | 50.2 ± 4.2 | 7.37 ± 0.03 | 7.32 ± 0.03[§] | 105 ± 10 | 109 ± 36 |

*Table 2 continued on next page*

*Table 2 continued*

| Baseline | Temperature (°C) Male | Female | P$_a$O$_2$ (mmHg) Male | Female | P$_a$CO$_2$ (mmHg) Male | Female | pH Male | Female | MAP (mmHg) Male | Female |
|---|---|---|---|---|---|---|---|---|---|---|
| Neonatal LPS + Keto + sAIH | 37.5 ± 0.2 | 37.4 ± 0.3 | 252 ± 24 | 245 ± 21 | 42.2 ± 3 | 51.2 ± 6.9 | 7.38 ± 0.02 | 7.31 ± 0.04 | 117 ± 14 | 125 ± 19 |
| Neonatal Saline + CGS-21680 | 37.3 ± 0.3 | 37.6 ± 0.1 | 270 ± 46 | 215 ± 48 | 45.7 ± 4.9 | 47 ± 3.3 | 7.34 ± 0.06 | 7.35 ± 0.04 | 112 ± 30 | 126 ± 21 |
| Neonatal LPS + CGS-21680 | 37.4 ± 0.4 | 37.4 ± 0.4 | 254 ± 26 | 256 ± 21 | 46.1 ± 3.2 | 46.5 ± 3.2 | 7.37 ± 0.01 | 7.33 ± 0.04 | 107 ± 16 | 101 ± 25 |
| Time Controls | 37.5 ± 0.3 | | 220 ± 25 | | 48.4 ± 3.7 | | 7.36 ± 0.04 | | 102 ± 22 | |
| Time Controls + Keto | 37.6 ± 0.2 | | 272 ± 21 | | 44.3 ± 8.7 | | 7.37 ± 0.07 | | 111 ± 48 | |
| CGS-21680 Vehicle Controls | 37.6 ± 0.2 | | 271 ± 27 | | 45.4 ± 1.6 | | 7.36 ± 0.03 | | 108 ± 7 | |

MAP, mean arterial pressure; P$_a$O$_2$, arterial oxygen pressure; P$_a$CO$_2$, arterial carbon dioxide pressure. Neonatal Saline +mAIH male (n = 7) female (n = 7); Neonatal LPS +mAIH male (n = 12) female (n = 6); Neonatal Saline +Keto + mAIH male (n = 4) female (n = 5); Neonatal LPS +Keto + mAIH male (n = 4) female (n = 5); Neonatal Saline +sAIH male (n = 5) female (n = 4); Neonatal LPS +sAIH male (n = 4) female (n = 4); Neonatal Saline +Keto + sAIH male (n = 5) female (n = 5); Neonatal LPS +Keto + sAIH male (n = 5) female (n = 6); Time Control (n = 5); Time Control + Keto (n = 4). Statistical comparisons: ANOVA-RM, Tukey's post hoc: [*] different from Time control within time point, [†] different from TC +Keto within time point, [‡] different from baseline and 60 min, [§] different from baseline, [#] different from female neonatal LPS +Keto + sAIH within time point, [¶] different from female LPS within time point

DOI: https://doi.org/10.7554/eLife.45399.021

The following source data is available for Table 2:
Source data 1. Physiological parameters.
DOI: https://doi.org/10.7554/eLife.45399.022

Q- and S-pathways. Thus, future studies are needed to understand how inflammation modifies cross-talk between Q- and S-pathways and how respiratory motor plasticity can be exploited therapeutically (*Gonzalez-Rothi et al., 2015*).

The timing of neonatal inflammation is likely a significant factor in how neonatal inflammation impacts adult physiology. Low-levels of cytokines are important for neurodevelopment (*Bilbo and Schwarz, 2009*), and perturbing the balance of neonatal cytokines during development leads to lasting aberrant effects on neural circuits and developing cells (*Reemst et al., 2016*). Furthermore, while many components of the respiratory system begin developing *in utero* (*Prakash et al., 2000*; *Pagliardini et al., 2003*; *Mantilla and Sieck, 2008*; *Johnson et al., 2018*), the respiratory control system undergoes significant postnatal maturation. In these studies, we induced systemic inflammation with LPS at P4, similar to other studies showing long-term consequences of neonatal inflammation in other physiological systems (*Shanks et al., 2000*; *Walker et al., 2006*; *Fan et al., 2008*; *Kohman et al., 2008*; *Bilbo, 2010*), supporting the idea that important neural changes occur within the first week of life. Yet, it remains to be determined whether there is a precise critical period where neonatal inflammation impacts respiratory control circuits. However, our data on male mortality after neonatal LPS are consistent with other critical developmental windows, including a male-specific sensitive period to LPS (*Rourke et al., 2016*), disproportionate male mortality from neonatal inflammation (*Person et al., 2014*), the increased risk of sudden infant death syndrome for males (*Kinney and Thach, 2009*), and increased incidence of obstructive sleep apnea in adults after neonatal inflammation (*McNamara and Sullivan, 2000*). Thus, these data have important implications for understanding the sex-specific impairment early in life and into adulthood. Additionally, we and others (*Spencer et al., 2006*) observed a short delay in weight gain after neonatal inflammation, which normalized by weaning, suggesting no lasting effects on growth. Future studies are needed to refine our understanding of the critical periods during development when early-life inflammation induces long-lasting physiological changes to improve our understanding of adult disease and better understand important developmental processes.

While other reports have shown sex differences in neonatal programming of adult neuro-inflammatory responses (*LaPrairie and Murphy, 2007*; *Rana et al., 2012*), we observed no sex-differences in the effects of neonatal inflammation on adult plasticity. Importantly, this is the first evidence of inflammation abolishing pLTF in females and the first to report sAIH-induced respiratory motor plasticity in females. Females exhibited greater acute hypoxic phrenic amplitude responses relative to

males, consistent with previous findings (*Mortola and Saiki, 1996*; *Bavis et al., 2004*), despite variability in reports of sex differences in hypoxic ventilatory responses (*Behan and Kinkead, 2011*). In contrast to our results following neonatal inflammation, neonatal stress alters adult hypoxic responses in a sex-dependent manner, whereby male responses are enhanced and female responses are blunted (*Rousseau et al., 2017*). Thus, the long-term effects on respiratory control may be dependent on the type of stressors in early life. Importantly, our experiments were performed in adult, ovariectomized females with exogenously restored estradiol levels to permit respiratory motor plasticity (*Behan et al., 2002*; *Zabka et al., 2003*; *Dougherty et al., 2017*). Therefore, as sex hormones are known to modulate respiratory control and hypoxic responses (*Nelson et al., 2011*; *Behan and Kinkead, 2011*), we cannot rule out a confounding role for exogenous estradiol supplementation after ovariectomy. Finally, after neonatal inflammation, we found no differences in adult hypoxic responses, suggesting no lasting change in carotid body responses due to neonatal inflammation. Accordingly, the deficit in adult respiratory motor plasticity after neonatal inflammation is likely a consequence of long-term changes in the spinal cord where pLTF occurs (*Baker-Herman et al., 2004*; *Devinney et al., 2015*; *Dale et al., 2017*).

While adult anti-inflammatory treatment restored Q-pathway-evoked pLTF, we did not observe increases in inflammatory gene expression in adult medullary or cervical spinal cord homogenates. Thus, while inflammatory signaling contributes to the impairment of adult plasticity, the source of this signaling change remains unclear and will be the topic of future studies. Similarly, others demonstrated no changes in baseline CNS inflammatory markers after neonatal inflammation, but observed priming of glial responses to adult stimuli (*Bilbo et al., 2005b*; *Mouihate et al., 2010*; *Smith et al., 2014*), suggesting lasting changes in glia have the potential to underlie impairments in adult respiratory plasticity. Contrary to other reports (*Boissé et al., 2005*; *Kentner et al., 2010*), we found spinal COX-2 gene expression was decreased in adulthood, suggesting a decrease in inflammatory signaling, which is unlikely to contribute to the lasting inflammation-dependent impairment in plasticity. Further, the acute inflammatory impairment of adult respiratory plasticity is COX-independent (*Huxtable et al., 2018a*), emphasizing a role for other inflammatory molecules mediating the lasting impairment in respiratory motor plasticity. Unmeasured inflammatory genes or post-transcriptional changes in inflammatory proteins may be responsible for undermining adult pLTF after neonatal inflammation. Conversely, other perinatal stimuli involving inflammatory signaling, such as maternal care and diet, do have lasting programming effects on adult inflammatory cytokine expression (*Bilbo and Schwarz, 2009*), but are more complex stimuli than the acute neonatal inflammation in our study. We also observed no change in microglial or astrocyte density and no obvious qualitative changes in morphology in adult medullas or spinal cords after neonatal inflammation. Thus, there are no obvious signs of inflammation in regions contributing to pLTF despite the restoration of Q-pathway-evoked pLTF with ketoprofen. Furthermore, the abolition of S-pathway-evoked pLTF is likely due to lasting changes in adenosine signaling from astrocytes, suggesting an astrocyte-specific change underlies this impairment. Thus, future studies are needed to identify inflammatory mechanisms undermining the Q-pathway and further details of the inflammation-independent mechanism responsible for undermining S-pathway-evoked motor plasticity.

The adult respiratory control network is vulnerable to early life stressors (*Bavis et al., 2004*; *Genest et al., 2004*; *Fournier et al., 2011*), which may undermine the ability to compensate during adult ventilatory control disorders. Our study is the first to demonstrate lasting consequences of neonatal inflammation on adult respiratory control. These deficits in respiratory control are independent of later life events, in contrast to other studies in which the physiological effects of early life inflammation are not revealed until after an adult stimulus (*Bilbo et al., 2005b*; *Bilbo, 2010*). We found a single episode of neonatal systemic inflammation induced lasting impairment of both Q- and S-pathway-evoked respiratory motor plasticity in adults. Our results suggest the adult impairment of Q-pathway plasticity is dependent on acute inflammatory signaling; however, we observed no lasting increase in adult inflammatory gene expression or the density of astrocytes and microglia. The pharmacological induction of S-pathway-evoked pLTF demonstrates phrenic motor neurons are capable of plasticity and suggest upstream impairment, such as the source of adenosine. While strong evidence supports astrocytes as the primary source of adenosine during hypoxia (*Takahashi et al., 2010*; *Angelova et al., 2015*), we cannot rule out other sources of adenosine. Identifying cell-type specific changes underlying lasting physiological impairments will be explored in future studies. Future studies will investigate the lasting effects of neonatal inflammation on

isolated microglia and astrocytes to uncover potential mechanisms of adult impairments after neonatal inflammation.

Together, these results indicate two mechanistic pathways to spinal motor plasticity induced by AIH are undermined by neonatal inflammation in rats. Our experimental approach assessed phrenic nerve output in anesthetized rats and may not be generalizable to respiratory control in freely behaving animals or to other forms of motor plasticity. However, AIH induces long-term facilitation of ventilation in humans (*Mateika and Komnenov, 2017*) and strengthens corticospinal pathways to non-respiratory motor-neurons (*Christiansen et al., 2018*), suggesting our results likely have relevance to mechanisms of human spinal motor plasticity after AIH. While AIH-induced respiratory motor plasticity does not necessarily alter normal homeostatic control of ventilation, the general facilitation of spinal motor output has significant therapeutic potential for treating patients with respiratory and non-respiratory motor limitations (*Trumbower et al., 2012*; *Trumbower et al., 2017*; *Nichols et al., 2013*; *Hayes et al., 2014*)

In conclusion, this basic science study has major implications for the understanding the neonatal origins of adult ventilatory control disorders. These studies are the first evidence that one neonatal inflammatory exposure induces long-term impairments in adult respiratory control with potential relevance to many respiratory disorders. These findings are particularly relevant since inflammation is common in neonates (*Person et al., 2014*), especially those born prematurely who are at higher risk for adult disease (*Luu et al., 2016*). Improving our appreciation of how early life inflammation can influence adult respiratory control will have important consequences for understanding adult disease and susceptibility to respiratory disorders. Additionally, AIH-induced spinal motor plasticity is also a promising therapy to enhance motor recovery after spinal injury (*Trumbower et al., 2012*). However, not all patients respond to AIH (*Hayes et al., 2014*; *Trumbower et al., 2017*) and our findings suggest neonatal inflammatory exposure could contribute to these therapeutic limitations and understanding the mechanisms undermining plasticity will increase the therapeutic potential of AIH-induced spinal motor plasticity.

# Materials and methods

**Key resources table**

| Reagent type (species) or resource | Designation | Source or reference | Identifiers | Additional information |
|---|---|---|---|---|
| Chemical compound, drug | LPS (Lipopolysaccharides from e Coli (0111:B4)) | Sigma Aldrich | L4130 | dissolved in saline, 1 mg/ml |
| Chemical compound, drug | Keto ((S) - (+) - Ketoprofen) | Sigma Aldrich | 471909 | dissolved in 50% ethanol in saline, 12.5 mg/ml |
| Chemical compound, drug | CGS-21680 | Sigma Aldrich | C141 | dissolved in DMSO to 50 mM for storage in aliquots. Dissolved to 100 uM in 10% DMSO and artificial CSF for injections. |
| Antibody | anti-GFAP (Rabbit polyclonal) | Millipore | (Millipore Cat# AB5804, RRID:AB_2109645) | (1:1000) |
| Antibody | anti-NK1R (Guinea pig polyclonal) | Millipore | (Millipore Cat# AB15810, RRID:AB_11213393) | (1:500) |
| Antibody | anti-IBA1 (Rabbit polyclonal) | Wako | (Wako Cat# 019–19741, RRID:AB_839504) | (1:1000) |
| Antibody | anti-CHaT (Goat polyclonal) | Millipore | (Millipore Cat# AB144P, RRID:AB_2079751) | (1:300) |

*Continued on next page*

*Continued*

| Reagent type (species) or resource | Designation | Source or reference | Identifiers | Additional information |
|---|---|---|---|---|
| Antibody | donkey-anti-rabbit 647 IgG secondary | Life Technologies | (Molecular Probes Cat# A-31573, RRID:AB_2536183) | (1:1000) |
| Antibody | donkey-anti-goat 555 IgG secondary | Life Technologies | (Molecular Probes Cat# A-21432, RRID:AB_141788) | (1:1000) |
| Antibody | donkey-anti-guinea pig 488 IgG secondary | Jackson Immuno | (Jackson ImmunoResearch Labs Cat# 706-545-148, RRID:AB_2340472) | (1:1000) |
| Sequence-based reagent | IL-1β forward primer | Integrated DNA Technologies | CTG CAG ATG CAA TGG AAA GA | |
| Sequence-based reagent | IL-1β reverse primer | Integrated DNA Technologies | TTG CTT CCA AGG CAG ACT TT | |
| Sequence-based reagent | IL-6 forward primer | Integrated DNA Technologies | GTG GCT AAG GAC AAA GAC CA | |
| Sequence-based reagent | IL-6 reverse primer | Integrated DNA Technologies | GGT TTG CCG AGT AGA CCT CA | |
| Sequence-based reagent | iNOS forward primer | Integrated DNA Technologies | AGG GAG TGT TGT TCC AGG TG | |
| Sequence-based reagent | iNOS reverse primer | Integrated DNA Technologies | TCT GCA GGA TGT CTT GAA CG | |
| Sequence-based reagent | TNFα forward primer | Integrated DNA Technologies | TCC ATG GCC CAG ACC CTC ACA C | |
| Sequence-based reagent | TNFα reverse primer | Integrated DNA Technologies | TCC GCT TGG TGG TTT GCT ACG | |
| Sequence-based reagent | COX2 forward primer | Integrated DNA Technologies | TGT TCC AAC CCA TGT CAA AA | |
| Sequence-based reagent | COX2 reverse primer | Integrated DNA Technologies | CGT AGA ATC CAG TCC GGG TA | |
| Sequence-based reagent | 18 s forward primer | Integrated DNA Technologies | CGG GTG CTC TTA GCT GAG TGT CCC | |
| Sequence-based reagent | 18 s reverse primer | Integrated DNA Technologies | CTC GGG CCT GCT TTG AAC AC | |

All experiments were approved by the Institutional Animal Care and Use Committees at the University of Oregon and the University of Wisconsin-Madison and conformed to the policies of the National Institute of Health *Guide for the Care and Use of Laboratory Animals*. Male and female Sprague Dawley rats (Envigo Colony 217 and 206) were housed under standard conditions (12:12 hr light/dark cycle) with food and water *ad libitum*.

## Drugs and materials

LPS (0111:B4, Sigma Chemical) was dissolved and sonicated in sterile saline for neonatal intraperitoneal (i.p.) injections (1 mg/kg). S-(+) Ketoprofen (Keto, Sigma Chemical) was dissolved in ethanol (50%) and sterile saline for acute, adult injections (12.5 mg/ml/kg, i.p., 3 hr). 17-$\beta$ estradiol was dissolved in sesame oil (Tex Lab Supply, Texas, USA) for acute injections (40 µg/mL/kg, i.p.,3 hr) in adult females after ovariectomy.

The adenosine 2A receptor agonist CGS-21680 was dissolved in fresh artificial cerebrospinal fluid (aCSF: 120 mM NaCl, 3 mM KCl, 2 mM CaCl$_2$, 2 mM MgCl$_2$, 23 mM NaHCO$_3$, and 10 mM glucose) with DMSO (10%) for intrathecal injections.

## Neonatal treatments

Timed pregnant rats (E14-17 upon arrival) were purchased in pairs from a commercial vendor (Envigo) and monitored daily. To control for between litter effects, litters were stratified such that each dam fostered similar numbers of male and female pups. On postnatal day 4 (P4), all of the stratified pups with one dam were injected with LPS (1 mg/kg, i.p.), while pups with the control dam were injected with sterile saline (i.p.). The dose of LPS was based on previous studies demonstrating CNS inflammatory gene expression in neonates (*Rourke et al., 2016*), as well as our unpublished data (N Morrison, S Johnson, J Watters, A Huxtable, unpublished observations) indicating CNS inflammation following LPS (1 mg/kg). Pups were weighed weekly and weaned at P21. Electrophysiology experiments were conducted once males reached 300 g. Females were ovariectomized at approximately 250 g, 7–8 days prior to electrophysiology experiments.

## Ovariectomy

Ovariectomies were performed as previously described (*Dougherty et al., 2017*) to control for the known effects of estrus cycle hormones on pLTF (*Zabka et al., 2001*; *Behan et al., 2002*; *Dougherty et al., 2017*). Adult rats were anesthetized with isoflurane and maintained on a nose cone (2.5% in $O_2$) during surgery. Depth of anesthesia was confirmed by the absence of toe-pinch responses. Bilateral dorsolateral incisions exposed ovarian fat pads. Ovaries were ligated and removed, muscle layers were approximated, and skin incisions were closed with a single dissolvable suture. A single dose of buprenorphine (0.05 g/kg, s.c.) was administered at the end of surgery for pain control and rats recovered in individual cages for 7-8 days before electrophysiology studies. Since pLTF exists in females only when estradiol is high (*Dougherty et al., 2017*), estradiol levels were restored by injection of 17-β estradiol (40 µg/mL/kg, i.p.) three hours before electrophysiology experiments.

## Experimental groups

All experimental groups consisted of adult male and female rats after a single injection of either neonatal LPS or neonatal saline. To investigate the impact of neonatal systemic inflammation on adult Q-pathway-evoked respiratory motor plasticity, the following experimental groups were used: male neonatal saline + mAIH (n = 7), male neonatal LPS + mAIH (n = 12), female neonatal saline + mAIH (n = 7), female neonatal LPS + mAIH (n = 6).

To investigate if acute anti-inflammatory treatment restores Q-pathway-evoked respiratory motor plasticity after neonatal inflammation, adults were treated with ketoprofen (12.5 mg/kg, i.p.) three hours before electrophysiology experiments: male neonatal saline + Keto + mAIH (n = 4), male neonatal LPS + Keto + mAIH (n = 4), female neonatal saline + Keto + mAIH (n = 5), female neonatal LPS + Keto + mAIH (n = 5).

To investigate the impact of neonatal systemic inflammation on adult S-pathway-evoked respiratory motor plasticity, we used the following experimental groups: male neonatal saline + sAIH (n = 5), male neonatal LPS + sAIH (n = 4), female neonatal saline + sAIH (n = 4), female neonatal LPS + sAIH (n = 4).

To investigate if acute anti-inflammatory treatment restores S-pathway-evoked respiratory motor plasticity after neonatal inflammation, adults were treated with ketoprofen (12.5 mg/kg, i.p.) three hours before electrophysiology experiments: female neonatal saline + Keto + sAIH (n = 5), male neonatal LPS + Keto + sAIH (n = 5), female neonatal saline + Keto + sAIH (n = 5), female neonatal LPS + Keto + sAIH (n = 6).

To investigate if intermittent, intrathecal CGS-21680 reveals S-pathway-evoked respiratory motor plasticity, we used the following experimental groups: male neonatal saline + CGS-21680 (n = 4), male neonatal LPS + CGS-21680 (n = 6), female neonatal saline + CGS-21680 (n = 4), female neonatal LPS + CGS-21680 (n = 6).

To reduce use of additional animals, and because time control experiments were not statistically different between males or females, time control groups consisted of animals from each experimental condition. The time control group for studies investigating the Q-pathway (*Figure 2*) and S-pathway (*Figure 5*) consisted of adults after neonatal saline (male: n = 1; female n = 2), neonatal LPS (n = 1 male, 1 female). The time control + Keto group (*Figures 3* and *6*) consisted of adults after neonatal saline + Keto (n = 1 male, 1 female) and neonatal LPS + Keto (n = 1 male, 1 female). Vehicle

controls for intrathecal CGS-21680 experiments (*Figure 7*) consisted of adults after neonatal saline (n = 1 male, 1 female) and neonatal LPS (n = 1 male, 1 female).

## Electrophysiological studies

Electrophysiological studies have been described in detail previously (*Bach and Mitchell, 1996*; *Baker-Herman and Mitchell, 2002*; *Huxtable et al., 2013*). Rats were anesthetized with isoflurane, tracheotomized, ventilated (Rat Ventilator, VetEquip), and vagotomized bilaterally. A venous catheter was placed for drug delivery and fluid replacement, and a femoral arterial catheter was used to monitor blood pressure and for arterial blood sampling. Arterial blood samples were analyzed ($PaO_2$, $PaCO_2$, pH, base excess; Siemens RAPIDLAB 248) during baseline, during the first hypoxic response, and 15, 30 and 60 min post-AIH. Temperature was measured with a rectal temperature probe (Kent Scientific Corporation) and maintained between 37°C and 38°C with a custom heated table. Using a dorsal approach, hypoglossal and phrenic nerves were cut distally, and de-sheathed. Rats were converted to urethane anesthesia (1.8 g/kg i.v.; Sigma-Aldrich), allowed to stabilize for one hour, and paralyzed with pancuronium dibromide (1 mg; Selleck Chemicals).

In rats receiving intrathecal injections, a laminectomy was performed at cervical vertebrae 2 (C2) and a primed, silicone catheter was inserted two millimeters through a small incision in the dura. The catheter tip extended toward the rostral margin of C4 (*Baker-Herman and Mitchell, 2002*). CGS-21680 (100 μM) or vehicle (10% DMSO in aCSF) was injected around the phrenic motor pool in three boluses (10 μL) separated by 5 min.

Nerves were bathed in mineral oil and placed on bipolar silver wire electrodes. Raw nerve recordings were amplified (10 k), filtered (0.1–5 kHz), integrated (50 ms time constant), and recorded (10 kHz sampling rate) for offline analysis (PowerLab and LabChart 8.0, AD Instruments). Apneic and recruitment $CO_2$ thresholds were determined by changing inspired $CO_2$ with continuous end-tidal $CO_2$ monitoring (Kent Scientific Corporation). End tidal $CO_2$ was set 2 mmHg above the recruitment threshold, whereby arterial blood samples were used to establish baseline $PaCO_2$, which was maintained within 1.5 mmHg of the baseline value throughout. Blood volume and base excess were maintained (±3 MEq/L) by continuous infusion (1–3 mL/h, i.v.) of hetastarch (0.3%) and sodium bicarbonate (0.99%) in lactated ringers. Experiments were excluded if mean arterial pressure deviated more than 20 mmHg from baseline.

All rats (excluding time control rats) received three, 5 min bouts of either mAIH (~10.5% $O_2$, $PaO_2$35–45 mmHg) or sAIH (~7% $O_2$, $PaO_2$25–35 mmHg). The average amplitude and frequency of 30 consecutive integrated phrenic bursts were taken during baseline, the first acute hypoxic response, and 15, 30, and 60 min after AIH and made relative to baseline amplitude. Phrenic nerve activity data for each experimental group were compared using two-way, repeated measures ANOVA with Fisher LSD post hoc tests. Sample sizes were selected based on similar, previous studies and the variance of pLTF in our experience (*Huxtable et al., 2018a*; *Huxtable et al., 2018b*; *Hocker and Huxtable, 2018*). Physiological variables were compared using two-way, repeated measures ANOVA with Tukey's post hoc test. Mean arterial pressure is reported for baseline, the end of the third hypoxic exposure, and 60 min after AIH. Acute hypoxic responses were compared using an ANOVA with Fisher LSD post hoc test. Values are means ± SD.

## RNA isolation, cDNA synthesis and quantitative PCR experiments

Neonatal rats (P4) were injected with either vehicle (saline) or LPS (1 mg/kg, i.p.) and allowed to mature to ~12 weeks. Adult male and female rats were anesthetized with isoflurane and perfused with PBS (transcardiac). Medulla and cervical spinal cords (C3-C7) were dissected and flash frozen until they were homogenized in Tri-Reagent (Sigma, St. Louis, MO, USA). Glycoblue reagent (Invitrogen, Carlsbad, CA, USA) was used to isolate total RNA, according to the manufacturer's protocol. cDNA was reverse transcribed from 1 μg of total RNA using MMLV reverse transcriptase together with a cocktail of oligo dT and random primers (Promega, Madison, WI, USA), as previously described (*Crain and Watters, 2015*), and analyzed using qPCR with PowerSYBR green PCR master mix on an ABI 7500 Fast system. Inflammatory gene expression was analyzed in medulla and spinal cord homogenates using the following primers:

IL-6: 5'-GTG GCT AAG GAC CAA GAC CA and 5'-GGT TTG CCG AGT AGA CCT CA;
IL-1β: 5'-CTG CAG ATG CAA TGG AAA GA and 5'-TTG CTT CCA AGG CAG ACT TT;

COX-2: 5'-TGT TCC AAC CCA TGT CAA AA and 5'-CGT AGA ATC CAG TCC GGG TA;
TNF-α: 5'-TCC ATG GCC CAG ACC CTC ACA C and 5'-TCC GCT TGG TGG TTT GCT ACG;
iNOS: 5'-AGG GAG TGT TGT TCC AGG TG and 5'-TCT GCA GGA TGT CTT GAA CG;
18 s: 5'-CGG GTG CTC TTA GCT GAG TGT CCC G and 5'-CTC GGG CCT GCT TTG AAC AC.

Wherever possible, primers were designed to span introns (Primer three software) and were purchased from Integrated DNA Technologies (Coralville, IA, USA). Primer efficiency was assessed by use of standard curves, as previously reported (Crain and Watters, 2015). Expression of inflammatory genes was made relative to 18 s ribosomal RNA calculated using the $2^{-\Delta\Delta CT}$ method (Livak and Schmittgen, 2001). Gene transcripts were considered undetectable, and not included in statistical analyses if their CT values fell outside of the linear range of the standard curve for that primer set, which in most cases was ≥34 cycles.

## Immunohistochemistry methods

Upon completion of electrophysiology experiments, rats were perfused (transcardiac) with cold phosphate buffered saline (PBS, pH 7.4), followed by 4% paraformaldehyde (pH 7.4). All brains were removed and immersed in paraformaldehyde until sectioning (Leica VT 1200S vibratome). For immunohistochemistry, transverse medullary and coronal cervical spinal cord sections (40 μm) were washed (PBS) and blocked (PBS, 0.3% Triton, 1% BSA, 2 hr, room temperature) to prevent non-specific antibody binding. For medullary sections, two combinations of primary antibodies were used (PBS, 0.3% Triton, 0.01% BSA, room temperature, 24 hr): (1) rabbit anti-GFAP (1:1000, Millipore AB5804) to label astrocytes and guinea pig anti-NK1R (1:500, Millipore AB15810) to label preBötzinger Complex (preBötC) neurons (Gray et al., 1999), and (2) rabbit anti-IBA1 (1:1000, Wako 019–19741) to label microglia and guinea pig anti-NK1R (1:500, Millipore AB15810) to label preBötC neurons. For the spinal cord, two different combinations of primary antibodies were used (PBS, 0.3% Triton, 0.01% BSA, room temperature, 24 hr): (1) rabbit anti-GFAP (1:1000, Millipore AB5804) to label astrocytes and goat anti-ChAT (1:300, Millipore AB144p) to label motor neurons, (2) rabbit anti-IBA1 (1:1000, Wako 019–19741) to label microglia and goat anti-ChAT (1:300, Millipore AB144p) to label motor neurons. After primary antibody incubation, sections were rinsed (PBS) and incubated with secondary antibodies (PBS, 0.3% triton, 0.01% BSA, room temperature, 3 hr): donkey-anti-rabbit 647 IgG (1:1000, Life Technologies A31573) to label GFAP and IBA1 primary antibodies, donkey-anti-goat 555 IgG (1:1000, Life Technologies A21432) to label ChAT primary antibody and donkey-anti-guinea pig 488 IgG (1:1000, Alexa Fluor 706-545-148) to label NK1R primary antibody. Sections were washed and mounted onto charged microscope slides, air dried and covered with prolong gold (Life technologies, P36930) to preserve the fluorescence. A glass cover slip was placed over the samples and sealed with clear nail polish. Slides were stored in the dark at 4°C until imaged. All immunohistochemistry experiments contained adult male and female tissues after neonatal saline (medulla: n = 5 males, seven females; spinal cord: n = 5 males, six females) or neonatal LPS (medulla: n = 6 males, four females; spinal cord: n = 6 males, three females).

## Image analysis methods

All immunofluorescent images (1024 × 1024 pixels, 40x magnification) were acquired using a Leica Microsystems CMS GmbH confocal microscope using the LAS X acquisition and viewing software (0.5 μm z-stack step increments). All images were taken using identical laser and gain settings and identically adjusted for contrast/brightness using ImageJ open source software to allow for comparisons across all groups. To quantify the density of microglia and astrocytes, maximum intensity projections for 20 μm of z-stacks from the medulla and cervical spinal cords were analyzed. Mean fluorescent intensity for each image within a single batch was made relative to the average fluorescent intensities of adults after neonatal saline samples within each sex (Paizs et al., 2009). Data are presented as percent change from adults after neonatal saline within each sex.

## Statistical analysis

GraphPad Prism 7.0 software was used for statistical analyses. Differences in mortality between treatments and between sexes was evaluated with Fisher's exact test. Phrenic nerve activity data for each experimental group were compared using two-way, repeated measures ANOVA with Fisher LSD *post hoc* tests. Physiological variables were compared using two-way, repeated measures

ANOVA with Tukey's *post hoc* test. Mean arterial pressure is reported from baseline, the end of the third hypoxic exposure, and 60 min after AIH. Acute hypoxic phrenic responses were compared using an ANOVA with Fisher LSD *post hoc* test. Microglial and astrocytic density comparisons were made between groups using a one-way ANOVA with multiple-comparisons *post hoc* tests. For all tests, $p < 0.05$ was considered significant and all data are expressed as mean ± SD.

## Additional information

### Funding

| Funder | Grant reference number | Author |
|---|---|---|
| National Institutes of Health | HL141249 | Adrianne G Huxtable |
| National Institutes of Health | HL111598 | Jyoti J Watters |

The funders had no role in study design, data collection and interpretation, or the decision to submit the work for publication.

### Author contributions
Austin D Hocker, Conceptualization, Data curation, Formal analysis, Investigation, Visualization, Methodology, Writing—original draft, Project administration, Writing—review and editing; Sarah A Beyeler, Investigation, Visualization, Writing—original draft; Alyssa N Gardner, Formal analysis, Investigation; Stephen M Johnson, Conceptualization, Resources, Investigation, Writing—review and editing; Jyoti J Watters, Conceptualization, Resources, Data curation, Funding acquisition, Project administration, Writing—review and editing; Adrianne G Huxtable, Conceptualization, Resources, Funding acquisition, Investigation, Methodology, Writing—original draft, Project administration, Writing—review and editing

### Author ORCIDs
Austin D Hocker http://orcid.org/0000-0002-2941-2581
Adrianne G Huxtable http://orcid.org/0000-0002-8745-2231

### Ethics
Animal experimentation: This study was performed in strict accordance with the recommendations in the Guide for the Care and Use of Laboratory Animals of the National Institutes of Health. All of the animals were handled according to approved institutional animal care and use committee protocols (#18-02) of the University of Oregon. All surgeries were performed under isoflurane or urethane anesthesia and every effort was made to minimize pain, distress, or discomfort.

### Decision letter and Author response
Decision letter https://doi.org/10.7554/eLife.45399.025
Author response https://doi.org/10.7554/eLife.45399.026

## Additional files

### Supplementary files
• Transparent reporting form
DOI: https://doi.org/10.7554/eLife.45399.023

### Data availability
All data generated or analyzed during this study are included in the manuscript and supporting files. Source data are available for all figures.

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
