## [Decision Letter]

[Editors’ note: a previous version of this study was rejected after peer review, but the authors submitted for reconsideration. The first decision letter after peer review is shown below.]

Thank you for submitting your work entitled "A single bout of neonatal inflammation impairs adult respiratory motor plasticity in male and female rats" for consideration by *eLife*. Your article has been reviewed by three peer reviewers, including Jan-Marino Ramirez as the Reviewing Editor and Reviewer #1, and the evaluation has been overseen by a Senior Editor.

Our decision has been reached after consultation between the reviewers. Based on these discussions and the individual reviews below, we regret to inform you that your work will not be considered further for publication in *eLife*.

The reviewers felt that additional experiments will be necessary to increase the significance of your findings. The reviewers had several specific recommendations and suggestions.

1) It would be important to demonstrate that there is indeed significant inflammation in areas relevant for LTF – e.g. phrenic nucleus – 48hrs after LPS injection and 1 or two weeks later.

2) The reviewers also recommended to add some experiments that show that LTF translates into an altered hypoxic ventilatory response in freely moving unanesthetized animal.

3) Another issue that was raised relates to the apparent lack of a glial involvement. It would be more convincing to the readers, if the authors would add a deeper glial characterization e.g. trough flow cytometry, an approach which you seem to be familiar with (Nikodemova et al., 2012; 2014; 2015; 2016), and/or an additional test to explore whether glial responsivity to inflammatory challenges has been altered (Li et al., 2010; Cai et al., 2013; Yin et al., 2013; Lunardelli et al., 2015). It is possible that the glia seem unchanged under "resting" conditions but become more prone (primed) towards activation, something you are also very familiar with (Huxtable et al., 2013; Johnson et al., 2018).

Reviewer #1:

This is a solid study by Hocker et al., investigating the consequences of neonatal LPS injection on respiratory plasticity. The authors report several interesting findings.

One finding is the increased neonatal mortality observed in male versus female pups. This is a very important finding in the context of SIDS, which affects males significantly more than female. The ratio of mortality seen in this study is similar to that seen in human SIDS. The only minor concern is that this interesting observation is not further explored. Hence it seems a bit of an "add-on" rather than an integral part of the story reported in this manuscript.

The remainder of the manuscript focuses on respiratory plasticity. It follows in the footsteps of the Gordon Mitchell laboratory and carefully dissects the so called "Longterm facilitation" into the different cellular pathways and the author characterize the underlying mechanisms of the LTF as well as the changes in female and males exposed to neonatal LPS.

The single bolus of LPS has long-term differential consequences on this form of plasticity and the underlying mechanisms. The authors perform e.g. a careful analysis of the cytokines involved – and the involvement is rather diverse. This is perhaps not surprising.

The one finding that seems unexpected is that the authors cannot identify the actual mechanism underlying the long-term impairment of the plasticity. They did not discover any changes in astrocytes and microglia, neither abundance of these cells nor in their morphology. This "negative finding" leaves the reader somewhat "unsatisfied" as it makes the study appear somewhat descriptive and "unresolved". However, this may not be the case – the authors have done a great job in unraveling the subcellular mechanisms underlying the plasticity and the changes associated with the one-time inflammatory event. Perhaps the lack of glial alterations is the most important finding, i.e. that the plasticity is disturbed in the absence of any obvious glial disturbance. In other words, the network assumed a different responsiveness and the glial environment has adjusted to this new state. Clinically, this seems like a very important finding.

Of course, the problem with any negative finding is that there might be changes that the authors did not discover using the methods applied.

Reviewer #2:

This manuscript describes experiments designed to determine the effect of 1 time exposure to LPS during neonatal period in newborn rats on respiratory motor plasticity in adulthood. The work comes from a well-established laboratory with significant contributions to our understanding of phrenic nerve plasticity measured as long-term facilitation after several short exposures to intermittent hypoxia. They have outlined the 2 distinct cellular and molecular pathways that underlies this plasticity (S and Q pathway). They describe in this manuscript the effect of sex on the outcome variables.

Introduction: The Introduction nicely outlines the rationale for the experiments and points out that little is known about the effect of sex on expression of pLTF after neonatal exposure to inflammation. The topic is important since there are several adult onset diseases that are likely secondary to early neonatal exposure to inflammation such as Parkinsonism, and pLTF is thought to be a protective reflex to sustain ventilation during hypoxic exposure. The hypothesis is clearly stated that neonatal inflammation undermines the Q-pathway but not the S-pathway, respiratory motor plasticity in adult male and female rats.

The major findings are that LPS exposure in neonatal rats (1) increased mortality only in male pup, (2) abolished the phrenic nerve plasticity (pLTF) induced by moderate intermittent hypoxia involving the Q-pathway, and acute exposure to ketoprofen given to adults restored the pLTF, (3) similarly, the S-pathway mediating pLTF was affected in both male and female adult rats but ketoprofen did not restore the response, lastly (4) adult females had a greater acute phrenic amplitude response during moderate and severe hypoxia but this was unaffected by LPS exposure or Keto. Of note, markers of inflammation (cytokine gene expression markers of microglia activation) examined was not seen in the areas of the brainstem and cervical spinal cord). The authors conclude the effects observed in adult animals suggest neonatal inflammation affects peripheral inflammatory signaling or protein expression that mediates the responses.

Overall the findings are interesting and the paper well written but very long. I do have some concerns about the conclusion stated by the authors based on the experimental design and the data presented. The following questions should be addressed.

1) The authors use the term "motor plasticity" but the experiments only measured neural activity from the phrenic nerve. They also refer to "ventilator responses" the animals are anesthestized, paralyzed and vagotimzed, they don't have a "ventilatory response". The authors should simply state what was measured the electrical responses of the phrenic neurogram.

2) Is it possible that these experiments could have been done in awake animals? Can you observe the long-term motor plasticity and true ventilatory responses? Experiments in awake animals would be more meaningful. How does the phrenic responses translate into the true response that we are interested in -Ventilation?

3) What were serum glucose levels during the experimental challenges?

4) Why did the male animals have greater mortality after LPS exposure?

5) Was a dose response of LPS done in preliminary experiments? If not, why was the current dose of LPS chosen?

6) Did LPS exposure ever elicit an inflammatory response within the brains of these animals during the neonatal period? This would have been of interest to even know that the amount given ever elicited a central response.

7) What "peripheral" systems do the authors believe the early LPS exposure is modifying.

8) Why did the authors choose ketoprofen vs other non-steroidal anti-inflammatory drugs or even steroids?

9) The authors also conclude that their data indicate that a persistent change in the adult inflammatory signaling contributes to the over findings since Ketoprofen restores the response. Does this imply that the S pathway is not the modified by inflammation? The authors never demonstrate that "any inflammation" (central or peripheral) exist in the animals What are the off target non-anti-inflammatory effects of ketoprofen? Is the difference between the effects of the ketoprofen on the Q and S pathway simply because a high enough dose wasn't used?

10) Tissues were examined after the acute experiment was performed. Were tissues from LPS and saline exposed animals examined prior to the acute hypoxic experiments? Is it possible that the acute exposures may have modified the gene and protein expression pattern?

11) The paper is quite long, and a lot of the findings show no difference between the groups; I think the figures could be reduced by 25%. There is also an excessive number of references.

12) Not sure that is necessary for the authors to reference so many papers from their laboratory. There are an excessive number of references.

Reviewer #3:

The study "A single bout of neonatal inflammation impairs adult respiratory motor plasticity in male and female rats" by Hocker et al., shows that a single application of Lipopolysaccharide (LPS) after birth produces a reduction of different forms of long-term facilitation (LTF), which can be reverted by ketoprofen, in some cases. The finding that early inflammation induces long-lasting effects on breathing plasticity agrees with a wide variety of findings showing that early inflammation induces long-lasting effects on a several brain circuits (Walker et al., 2003; Spencer et al., 2006; Li et al., 2010; Kentner et al., 2010; Kirsten et al., 2011; Fan et al., 2011; Wang et al., 2011; Cai et al., 2013; Wang et al., 2013; Rideau Batista Novais et al., 2014; Lan et al., 2015; Comim et al., 2016; Patil et al., 2016; Onufriev et al., 2017; Singh et al., 2017; Réus et al., 2017; including the spinal cord; Walker et al., 2003; Li et al., 2010) and their plasticity (Walker et al., 2003; Rideau Batista Novais et al., 2014; Onufriev et al., 2017). Surprisingly, in contrast with other reports, the effects observed in this study did not correlate with changes in glia morphology (for contrast see: Fan et al., 2011; Wang et al., 2011; Cai et al., 2013; Wang et al., 2013; Yin et al., 2013; Pang et al., 2013; Lan et al., 2015) and the expression of pro-inflammatory genes (for contrast see: Cai et al., 2013; Yin et al., 2013; Patil et al., 2016, Onufriev et al., 2017) or proinflammatory mediators (for contrast see: Spencer et al., 2006; Kentner et al., 2010; Fan et al., 2011; Wang et al., 2013; Cai et al., 2013; Pang et al., 2013; Lunardelli et al., 2015; Lan et al., 2015; Patil et al., 2016; Onufriev et al., 2017; Réus et al., 2017). I consider the findings to be relevant, but also partially similar to the already described changes in other circuits of the brain by the same experimental manipulation (Walker et al., 2003;Spencer et al., 2006; Li et al., 2010; Kentner et al., 2010; Kirsten et al., 2011; Fan et al., 2011; Wang et al., 2011; Cai et al., 2013; Wang et al., 2013; Rideau Batista Novais et al., 2014; Lan et al., 2015; Comim et al., 2016; Patil et al., 2016; Onufriev et al., 2017; Singh et al., 2017; Réus et al., 2017). On the other hand, the mechanisms behind the findings of the present study are still foggy, since it is not clear what is the source of the inflammatory and the non-inflammatory signals involved in the described effects. Considering the contradictory evidence already available (Walker et al., 2003;Spencer et al., 2006; Li et al., 2010; Kentner et al., 2010; Kirsten et al., 2011; Fan et al., 2011; Wang et al., 2011; Cai et al., 2013; Wang et al., 2013; Rideau Batista Novais et al., 2014; Lan et al., 2015; Comim et al., 2016; Patil et al., 2016; Onufriev et al., 2017; Singh et al., 2017; Réus et al., 2017), the clarification of the mechanisms involved in the modulation described in this work would contribute to the relevance of the study.

[Editors’ note: what now follows is the decision letter after the authors submitted for further consideration.]

Thank you for submitting your article "One bout of neonatal inflammation impairs adult respiratory motor plasticity in male and female rats" for consideration by *eLife*. Your article has been reviewed by three peer reviewers, including Jan-Marino Ramirez as the Reviewing Editor and Reviewer #1 and the evaluation has been overseen by a Reviewing Editor and Ronald Calabrese as the Senior Editor.

The reviewers have discussed the reviews with one another and the Reviewing Editor has drafted this decision to help you prepare a revised submission.

Summary:

The manuscript by Hocker et al., has important implications. Many babies are exposed to inflammatory events early during their development, and due to an improved healthcare, these babies mostly survive and recover. Yet, it is unclear whether this recovery is complete, and whether even in the absence of any acute inflammation there are long-term consequences on respiratory control. The present study addresses this important issue employing an established rodent model that uses respiratory plasticity in the phrenic nerve as a well-defined respiratory output.

This manuscript is a resubmission of a previously reviewed manuscript, and the authors performed additional experiments to address several of the reviewers' original comments. Their new experiments shed new light on the potential involvement of astrocytes (as it relates to the S-pathway in case of severe AIH). The authors also added new experimental data indicating distinct differences between the impairment of Q and S-pathways caused by neonatal inflammation. These are major revisions to the previous manuscript – which is particularly remarkable since such developmental studies consume considerable time.

The authors also added a discussion of the importance of acute intermittent hypoxia in the context of spinal cord injury and the therapeutic application for motor neuron disease. The authors emphasize the important implication in particular that the application of AIH could be undermined by early inflammatory experiences.

The authors also included a table from a separate study that describes in detail the inflammatory gene data that are associated with the LPS induced inflammation. However, it makes sense that these data are not included in the present study – because they are published elsewhere. However, these data were helpful for evaluating the findings of the present study, because their careful characterization confirms that there is impairment of respiratory activity even though the inflammatory gene responses have returned to baseline.

This leads to the important conclusion that the observed loss of plasticity is unlikely due to an ongoing inflammation.

Essential revisions:

The reviewers concluded that no additional experiments will be required. However, some concerns remained unaddressed, and need to be carefully addressed in the revision.

1) The authors should include a limitation section that discusses the experimental approach: in particular whether their findings can be translated to a fully integrated/ intact model animal or human. Regardless of how the model might allow one to dissect out pathways that underlie pLTF, if the whole purpose of the reflex is to maintain ventilation to compensate for hypoxia, then it is still important to demonstrate that the findings are relevant to ventilation in intact, unanesthetized animals. Please address what the findings mean behaviorally, if they are transmitted to the muscles of respiration?

2) We suggest reducing the number of figures with negative findings, and instead include a table showing what is different between the groups, which pathway is implicated, and the gender effects. The table would allow the reader to get a better conceptual view of all the many variables that are being assessed in this study.

3) The Abstract is very short and not very informative. The age of the animal at the time of the exposure and the age of the animals when the pLTF was assessed is not included. Moreover, there are sex differences that also are not specified in the Abstract.

4) In their response, the authors have clearly indicated the novelty of their relevant findings, in contrast to the wealth of evidence regarding the long-term consequences of early-life inflammation. Which, by the way, was clear in the original submission. However, we think that it is still necessary to carefully deal with the fact that among those published effects there is a long-lasting change in astrocytes and microglia as well as in their inflammatory mediators. Since these changes were not observed in the submitted study, the authors should include a proper discussion of the source of these differences and they should discuss the alternative mechanisms that could potentially be involved. The authors have shown several aspects that are not involved in their novel findings, which we agree is very important. The authors have also included a set of experiments that indirectly indicate that the astrocytic function could be affected by early-life inflammation, but this needs to be carefully discussed. We agree with the authors that although indirect, these experiments hint that astrocytes could have been changed by early-life inflammation. What we would like to see in the revision is that the authors discuss future experiments or avenues that could more directly address their observations.

---

## [Author Response]

[Editors’ note: a previous version of this study was rejected after peer review, but the authors submitted for reconsideration. The first decision letter after peer review is shown below.]

Our decision has been reached after consultation between the reviewers. Based on these discussions and the individual reviews below, we regret to inform you that your work will not be considered further for publication in eLife.

We thank the reviewers for their recommendations, and have completed additional experiments to highlight distinct, mechanistic differences between Q- and S-pathway impairment after neonatal inflammation. We also now include additional evidence demonstrating that phrenic motor neurons are not impaired after neonatal inflammation, suggesting impairment in glial cell (e.g. astrocyte) function. Given the developmental nature of this study, collection and analysis of new data represent significant changes in the manuscript from the original submission and significantly enhance the overall impact of the manuscript. Lastly, we modified the manuscript to emphasize the clinical relevance of these studies by discussing the use of acute intermittent hypoxia (AIH)-induced plasticity therapeutically after spinal cord injury, and potential use for treatment of motor neuron disease. Our novel findings demonstrate that such therapeutic approaches would be undermined by experiences early in life.

The reviewers felt that additional experiments will be necessary to increase the significance of your findings. The reviewers had several specific recommendations and suggestions.1) It would be important to demonstrate that there is indeed significant inflammation in areas relevant for LTF – e.g. phrenic nucleus – 48hours after LPS injection and 1 or two weeks later.

Thank you for raising this issue, as we are also interested in the inflammatory state of CNS regions relevant to respiratory plasticity after LPS. The ability of LPS to induce inflammation at a variety of doses are well documented both in neonates and adults in many CNS regions (reviewed in Bilbo and Schwarz, 2012). Indeed, we have gone to considerable effort to characterize the effects of LPS-induced inflammation in respiratory control regions after neonatal LPS. In the context of this manuscript, the most salient point is that in adults, long after the inflammatory gene response has returned to baseline levels after neonatal LPS challenge, there are lasting impairments in respiratory activity. Figure 4 in the manuscript shows that there is no significant difference in inflammatory gene expression in adults challenged with either neonatal saline or LPS, despite the loss of respiratory motor plasticity in the LPS-challenged adults. Thus, while neonatal inflammation causes transient increases in the expression of inflammatory genes, the lasting effects of neonatal inflammation into adulthood either result from cell-type specific inflammatory signaling changes not captured in our analysis, or ongoing inflammation in other CNS regions impacting spinal motor plasticity.

2) The reviewers also recommended to add some experiments that show that LTF translates into an altered hypoxic ventilatory response in freely moving unanesthetized animal.

We agree that changes in ventilatory responses in freely behaving animals are of interest, and those studies represent the next stage of experiments in this project. The present study is the first to characterize lasting impairments in the neural control of breathing after a single inflammatory insult during the neonatal period, and our current electrophysiologic approach has important strengths not afforded in freely behaving animals: (1) it enables measurement of changes in neural control independent of the mechanics of breathing, and (2) it allows us to better investigate mechanisms of respiratory motor plasticity using direct application of drugs onto the cervical spinal cord, where LTF occurs (see response to #3 above). While it is possible to measure respiratory motor plasticity in freely behaving animals (e.g. ventilatory LTF, see response to reviewer 2 #2), our interest is in the change in neural output caused by neonatal inflammation. Neuroplasticity cannot be studied in the context of whole animal breathing, and the powerful electrophysiologic model we have used here has allowed the identification of several cellular mechanisms underlying AIH-induced pLTF (reviewed in Turner et al., 2018). Further, studying spinal motor plasticity induced by acute intermittent hypoxia is of particular clinical relevance because intermittent hypoxia not only enhances respiratory motor output, but it also facilitates motor output throughout the spinal cord, and is used therapeutically to improve locomotor function in patients after spinal cord injury (Trumbower et al., 2012, 2017; Hayes et al., 2014). Thus, the impact of early life inflammation on motor neuron plasticity has clinical relevance for the utility of acute intermittent hypoxia as a therapeutic in regions independent of ventilation, underscoring the importance of studying plasticity in its own right. The restoration of spinal motor plasticity with acute anti-inflammatories suggests that anti-inflammatories may enhance the therapeutic benefit of acute intermittent hypoxia on motor output. We have revised the discussion accordingly, to include this important possibility (Discussion section).

3) Another issue that was raised relates to the apparent lack of a glial involvement. It would be more convincing to the readers, if the authors would add a deeper glial characterization e.g. trough flow cytometry, an approach which you seem to be familiar with (Nikodemova et al., 2012; 2014; 2015; 2016), and/or an additional test to explore whether glial responsivity to inflammatory challenges has been altered (Li et al., 2010; Cai et al., 2013; Yin et al., 2013; Lunardelli et al., 2015). It is possible that the glia seem unchanged under "resting" conditions but become more prone (primed) towards activation, something you are also very familiar with (Huxtable et al., 2013; Johnson et al., 2018).

We agree that it is possible that glia appear morphologically normal in adults, but that they are “primed” to respond differently after a subsequent adult stimulus. It is however important to highlight that the deficits in respiratory control presented here are independent of later life inflammatory events – findings that contrast with other studies showing the physiological effects of early life inflammation are only revealed after a subsequent adult stimulus (Bilbo and Schwarz, 2009; Wang et al., 2010; Bilbo, 2010; Kirsten et al., 2010, 2011; Roughton et al., 2013). That respiratory plasticity is impaired in the absence of a subsequent challenge underscores the novelty of our study and emphasizes the profound sensitivity of the neonatal respiratory circuitry to a single bout of early-life inflammation.

Nonetheless, based on the reviewer comments, we have now added an additional set of experiments that advance our understanding of the mechanistic underpinnings of, and glial involvement in, the impairment of S-pathway-evoked pLTF. The S-pathway is adenosine dependent (Nichols et al., 2012) and astrocytes are the primary source of adenosine during hypoxia (Takahashi et al., 2010; Angelova et al., 2015). Thus, we tested the hypothesis that astrocyte-phrenic motor neuron signaling is impaired in adults after neonatal inflammation. To test this idea, we utilized what is known about the mechanisms of S-pathway-evoked plasticity (Golder et al., 2008; Nichols et al., 2012), and episodically applied an adenosine receptor agonist. Our new data demonstrate that phrenic motor neurons are not generally impaired after neonatal inflammation as adenosine was able to elicit S-pathway-evoked pLTF. Rather, these new data suggest that the lasting impairment in S-pathway-evoked pLTF may be due to an alteration in adult spinal astrocytes, since the likely source of adenosine during severe intermittent hypoxia (sAIH) is astrocytes (Takahashi et al., 2010; Angelova et al., 2015). These results do not, however, rule out a role for altered microglia-astrocyte-neuron signaling, and the role of glia continue to be a topic of ongoing studies in our group. We are presently pursuing flow cytometry and RNAseq to better understand the status of astrocytes (and microglia) in adults following neonatal inflammation; these complex molecular analyses are beyond the scope of the present study.

Reviewer #1:This is a solid study by Hocker et al., investigating the consequences of neonatal LPS injection on respiratory plasticity. The authors report several interesting findings.One finding is the increased neonatal mortality observed in male versus female pups. This is a very important finding in the context of SIDS, which affects males significantly more than female. The ratio of mortality seen in this study is similar to that seen in human SIDS. The only minor concern is that this interesting observation is not further explored. Hence it seems a bit of an "add-on" rather than an integral part of the story reported in this manuscript.

Thank you for pointing out this important omission. We have now added the sentence below to our Discussion section to further highlight the importance of this finding. We are very interested in the sex differences in adult respiratory plasticity after neonatal inflammation, and studies are planned to further dissect their underlying mechanisms. Please also refer to response to reviewer 2 #4.

“Neonatal inflammation also increases male mortality consistent with clinical male mortality after neonatal inflammation (Person et al., 2014) and relevant to the increased risk of sudden infant death syndrome for males (Kinney and Thach, 2009).”

The remainder of the manuscript focuses on respiratory plasticity. It follows in the footsteps of the Gordon Mitchell laboratory and carefully dissects the so called "Long-term facilitation" into the different cellular pathways and the author characterize the underlying mechanisms of the LTF as well as the changes in female and males exposed to neonatal LPS.The single bolus of LPS has long-term differential consequences on this form of plasticity and the underlying mechanisms. The authors perform e.g. a careful analysis of the cytokines involved – and the involvement is rather diverse. This is perhaps not surprising.The one finding that seems unexpected is that the authors cannot identify the actual mechanism underlying the long-term impairment of the plasticity. They did not discover any changes in astrocytes and microglia, neither abundance of these cells nor in their morphology. This "negative finding" leaves the reader somewhat "unsatisfied" as it makes the study appear somewhat descriptive and "unresolved". However, this may not be the case – the authors have done a great job in unraveling the subcellular mechanisms underlying the plasticity and the changes associated with the one-time inflammatory event. Perhaps the lack of glial alterations is the most important finding, i.e. that the plasticity is disturbed in the absence of any obvious glial disturbance. In other words, the network assumed a different responsiveness and the glial environment has adjusted to this new state. Clinically, this seems like a very important finding.Of course, the problem with any negative finding is that there might be changes that the authors did not discover using the methods applied.

The authors thank the reviewer for their positive comments and careful consideration of the importance of a “negative finding”. We agree that the results are both intriguing and surprising (and also perhaps a bit frustrating!). However, as with any complex physiological process, it can be challenging at first to unravel all of the mechanisms underlying a phenomenon. As described above (response to #3 above), we have undertaken further studies to better understand the mechanisms underlying abolition of plasticity. Our results indicate adult impairment of the Q-pathway is a result of an inflammation-dependent process, and we are performing a number of molecular analyses to identify the molecule(s) key to this impairment. However, the inflammatory response is complex, and identifying the “culprit” inflammatory molecule will require extensive molecular analyses. Such studies represent the next several years of our research plan, and are thus, beyond the scope of the current manuscript.

Impairment of the S-pathway, on the other hand, appears to be independent of inflammatory signaling, and our new data (Figure 7 in the manuscript) demonstrate phrenic motor neurons are not impaired after neonatal inflammation. We propose that S-pathway impairment is due to a lasting impairment in adenosine signaling, the source of which (astrocytes) is likely altered by neonatal inflammation. Intermittent application of adenosine induces S-pathway plasticity by activating adenosine 2A receptors on phrenic motor neurons (Seven et al., 2017), a pathway that appears normal after neonatal inflammation, suggesting that phrenic motor neurons themselves do not exhibit lasting detriments in motor output after neonatal inflammation. We have restructured the order of the results to clarify the mechanistic results in this manuscript.

Reviewer #2:This manuscript describes experiments designed to determine the effect of 1 time exposure to LPS during neonatal period in newborn rats on respiratory motor plasticity in adulthood. The work comes from a well-established laboratory with significant contributions to our understanding of phrenic nerve plasticity measured as long-term facilitation after several short exposures to intermittent hypoxia. They have outlined the 2 distinct cellular and molecular pathways that underlies this plasticity (S and Q pathway). They describe in this manuscript the effect of sex on the outcome variables.Introduction: The Introduction nicely outlines the rationale for the experiments and points out that little is known about the effect of sex on expression of pLTF after neonatal exposure to inflammation. The topic is important since there are several adult onset diseases that are likely secondary to early neonatal exposure to inflammation such as Parkinsonism, and pLTF is thought to be a protective reflex to sustain ventilation during hypoxic exposure. The hypothesis is clearly stated that neonatal inflammation undermines the Q-pathway but not the S-pathway, respiratory motor plasticity in adult male and female rats.The major findings are that LPS exposure in neonatal rats (1) increased mortality only in male pup, (2) abolished the phrenic nerve plasticity (pLTF) induced by moderate intermittent hypoxia involving the Q-pathway, and acute exposure to ketoprofen given to adults restored the pLTF, (3) similarly, the S-pathway mediating pLTF was affected in both male and female adult rats but ketoprofen did not restore the response, lastly (4) adult females had a greater acute phrenic amplitude response during moderate and severe hypoxia but this was unaffected by LPS exposure or Keto. Of note, markers of inflammation (cytokine gene expression markers of microglia activation) examined was not seen in the areas of the brainstem and cervical spinal cord). The authors conclude the effects observed in adult animals suggest neonatal inflammation affects peripheral inflammatory signaling or protein expression that mediates the responses.Overall the findings are interesting and the paper well written but very long. I do have some concerns about the conclusion stated by the authors based on the experimental design and the data presented. The following questions should be addressed.1) The authors use the term "motor plasticity" but the experiments only measured neural activity from the phrenic nerve. They also refer to "ventilator responses" the animals are anesthestized, paralyzed and vagotimzed, they don't have a "ventilatory response". The authors should simply state what was measured the electrical responses of the phrenic neurogram.

Thank you for this comment and correction. Yes, we do use “motor plasticity” to describe LTF, as is common practice in studies using this model (Tadjalli et al., 2010; Agosto-Marlin et al., 2017; Lui et al., 2018; Seven et al., 2018). The manuscript has now been revised and the reference to ventilatory responses has been removed.

2) Is it possible that these experiments could have been done in awake animals? Can you observe the long-term motor plasticity and true ventilatory responses? Experiments in awake animals would be more meaningful. How does the phrenic responses translate into the true response that we are interested in -Ventilation?

The reviewer is correct in that aspects of the experiments here could be conducted in awake, freely behaving animals. For example, ventilatory LTF (vLTF), a lasting increase in ventilation after intermittent hypoxia in awake animals, can be used to monitor neuroplastic changes in breathing. However, a number of important limitations hinder the utility of using vLTF as a model for understanding respiratory neuroplasticity; the magnitude of vLTF is small (Olson et al., 2001), sleep state rapidly alters vLTF (Nakamura et al., 2010), and vLTF is much more variable than phrenic LTF (Olson et al., 2001). Furthermore, pLTF is a more useful model for understanding mechanistic aspects of respiratory plasticity, as the mechanistic underpinnings of pLTF have been studied by us and others for more than 25 years (Turner et al., 2018). Thus, we chose the most commonly studied model of respiratory plasticity, pLTF, to further probe mechanisms undermining adult plasticity after neonatal inflammation (see also response to #2 above). As exemplified in our newly included data (Figure 7) using this model, we demonstrate there is likely an astrocytic, adenosine-dependent mechanism disrupted by neonatal inflammation, which is responsible for undermining adult plasticity.

3) What were serum glucose levels during the experimental challenges?

In recent experiments, we have measured blood glucose levels at the beginning and end of phrenic nerve recordings. Due to the use of the anesthetic urethane (Wang et al., 2000), blood glucose levels are elevated in all adults regardless of whether they received neonatal saline or LPS. Importantly, adult blood glucose levels do not differ between adults that received neonatal saline or LPS (p = 0.2739), and they do not change over the course of electrophysiology recordings.

P4 Saline = 225 +/- 38 mg/dL, n=6 (4 males, 2 females) P4 LPS = 251 +/- 27 mg/dL, n=4 (2 males, 2 females)

4) Why did the male animals have greater mortality after LPS exposure?

The reviewer highlights an interesting aspect of our current study. The increase in male mortality is consistent with other reports of increased male vulnerability early in life. Interestingly, sex differences exist early in brain development to either establish or eliminate sex differences in brain function and behavior. Sex differences in microglial colonization exist at P4 in the hippocampus, parietal cortex, and amygdala with male rats having significantly more microglia than females (Bilbo and Schwarz, 2012; Schwarz et al., 2012). These sex differences have been proposed to explain different susceptibilities to certain developmental psychiatric disorders (Bilbo and Schwarz, 2012; Nelson and Guyer, 2012; Schwarz et al., 2012). Whether such differences also exist in respiratory-related regions are unknown but are currently under investigation in our laboratory. We chose not to speculate on the reason(s) for this increased mortality in males until we had more data to support such suggestions.

Although we failed to see any persistent differences in glial number or gross morphology in our current experiments, we and others, have hypothesized persistent differences in glial activities, and thus their behavior, might underlie increased male vulnerability (Nelson and Guyer, 2012; Reemst et al., 2016). Our understanding of neonatal astrocyte distribution and migration remains incomplete, but consensus is that astrogenesis continues postnatally. Both astrocytes and microglia have important roles in neuronal survival and programmed cell death. Synaptogenesis peaks postnatally and corresponds with astrogenesis. Further, glial dysfunction has been linked to numerous neurodevelopmental disorders, including: autism spectrum disorder (Zhan et al., 2014; Squarzoni et al., 2014), schizophrenia (Ma et al., 2013; Zhan et al., 2014), Fragile-X syndrome (Jacobs et al., 2010; Higashimori et al., 2013), depression (Nishiyama et al., 2002), and obsessive compulsive disorder (Zhan et al., 2014). Glial dysfunction has also been linked to Rett syndrome (Maezawa et al., 2009), which manifests with respiratory deficits such as irregular breathing, hyperventilation and apneas (Katz et al., 2009; Ren et al., 2012; Ramirez et al., 2013). Thus, because we are only in the beginning stages of characterizing sex differences in neural development which may underlie increased male mortality, this remains an area of active investigation in our laboratory and others’. Should the reviewers feel this warrants additional discussion in the manuscript, we are happy to include it; but we have remained cognizant of increasing the length of the manuscript given that the reviewers’ felt it was already too long.

5) Was a dose response of LPS done in preliminary experiments? If not, why was the current dose of LPS chosen?

We chose 1mg/kg LPS in our study based on our unpublished dose-response experiments. We found this to be the best dose where both neonatal mortality was minimal and inflammatory gene expression was transiently increased in both the neonatal medulla and cervical spinal cord, key respiratory control regions (see response 1 above for gene expression data). Our observations are also in line with inflammatory gene expression in other reports in neonates in both the medulla and spinal cord after 1 mg/kg LPS (Wang et al., 2006; Balan et al., 2011; Schwarz and Bilbo, 2011; Jafri et al., 2013), and are consistent with other experiments showing that higher LPS doses induce significant neonatal mortality (Blood-Siegfried et al., 2002; Rourke et al., 2016).

6) Did LPS exposure ever elicit an inflammatory response within the brains of these animals during the neonatal period? This would have been of interest to even know that the amount given ever elicited a central response.

Yes, LPS did initiate CNS inflammation in the neonate, and specifically in the medulla and cervical spinal cord (see response #1 above).

7) What "peripheral" systems do the authors believe the early LPS exposure is modifying.

We regret the lack of clarity on this point, and have revised the discussion accordingly. Since the anti-inflammatory ketoprofen restored Q-pathway-evoked plasticity even in the absence of detectable CNS inflammation in adults, our results suggest inflammatory signaling (either central and/or peripheral) contributes to plasticity deficits in adults after neonatal inflammation. We hypothesize that other inflammatory molecules not evaluated here, or inflammation in other important brain regions or peripheral tissues may be responsible for the lasting ketoprofen-sensitive inflammatory signaling undermining adult Q-pathway-evoked plasticity. RNA-Seq analyses are in progress to evaluate all inflammatory genes expressed.

“Additionally, no obvious differences in astrocyte or microglial morphology in adult phrenic motor nuclei or the preBötC were seen following neonatal LPS inflammation, suggesting other inflammatory mechanisms may be responsible for impairing adult pLTF.” Results section.

8) Why did the authors choose ketoprofen vs other non-steroidal anti-inflammatory drugs or even steroids?

Ketoprofen has previously been used to restore respiratory motor plasticity after acute inflammatory stimuli in adult rats (Huxtable et al., 2013, Huxtable et al., 2015). Ketoprofen is a non-selective NSAID targeting both COX-1 and COX-2, as well as NF-κB at the high doses used here (Cashman, 1996, Yin et al., 1998). Since acute inflammatory impairment of LTF in adults is COX-independent (Huxtable et al., 2018), we suggest that ketoprofen restores LTF either by inhibiting NF-κB activation (and thus subsequent inflammatory gene activation) or by other COX-independent mechanisms. Glucocorticoids, on the other hand, are well-known to suppress BDNF expression, a molecule critical for Q-pathway-evoked pLTF. Thus, the use of steroidal anti-inflammatories would have precluded our ability to accurately interpret the results.

9) The authors also conclude that their data indicate that a persistent change in the adult inflammatory signaling contributes to the over findings since Ketoprofen restores the response. Does this imply that the S pathway is not the modified by inflammation? The authors never demonstrate that "any inflammation" (central or peripheral) exist in the animals What are the off target non-anti-inflammatory effects of ketoprofen? Is the difference between the effects of the ketoprofen on the Q and S pathway simply because a high enough dose wasn't used?

Please also see response to reviewer 2 #8 above. As discussed in our previous publications (Huxtable et al., 2015; Huxtable et al., 2018), ketoprofen is a potent and common anti-inflammatory and analgesic agent used in many species, including humans (Foster et al., 1988) and rats (Cabre et al., 1998). The dose of ketoprofen used here is nearly 8x the median effective dose in rats (Cabré et al., 1998), so based on drug solubility and toxicity studies in other reports, higher doses are not feasible. The acute inflammation-independent nature of the S-pathway has been previously reported (Agosto-Marlin et al., 2017), so it follows that blocking acute inflammatory signaling in adults after neonatal inflammation would not restore LTF. We do not think the differential sensitivity of the Q and S pathways to inflammation are dependent on the ketoprofen dose, but rather, the lasting deficit in S-pathway plasticity is due to an astrocytic, adenosine-dependent mechanism (see response to reviewer 1).

10) Tissues were examined after the acute experiment was performed. Were tissues from LPS and saline exposed animals examined prior to the acute hypoxic experiments? Is it possible that the acute exposures may have modified the gene and protein expression pattern?

Tissues taken for gene expression analysis were taken from separate animals not used for neurophysiology experiments. However, tissues for immunohistochemistry experiments were taken at the end of the acute neurophysiology experiment because previous experiments by our group have shown no significant differences in inflammatory gene expression after AIH in CNS tissues (Smith et al., 2013). This practice is routine in our lab to reduce the total number of animals used. The methods now correctly reflect how tissues were used.

11) The paper is quite long, and a lot of the findings show no difference between the groups; I think the figures could be reduced by 25%. There is also an excessive number of references.

We agree that there is a substantial amount of data in this manuscript. However, due to the complexity of groups (neonatal inflammation/saline, age, sex, and other treatments), we separated the figures based on the individual hypotheses being tested to reduce density in the figures and improve clarity. If the reviewer has suggestions regarding which figures could be combined or eliminated, we would be happy to re-evaluate.

12) Not sure that is necessary for the authors to reference so many papers from their laboratory. There are an excessive number of references.

The references to our previous papers appear most in the Introduction and Materials and methods sections and were cited because they laid the foundation for the present study by establishing the acute effects of inflammation in adults. Additionally, much of the methodology used here has previously been used by our group, so to reduce reiterating previously published work, we instead direct readers to the previously published work. However, we have now removed redundant references in the Introduction and Discussion section.

Reviewer #3:The study "A single bout of neonatal inflammation impairs adult respiratory motor plasticity in male and female rats" by Hocker et al,. shows that a single application of Lipopolysaccharide (LPS) after birth produces a reduction of different forms of long-term facilitation (LTF), which can be reverted by ketoprofen, in some cases. The finding that early inflammation induces long-lasting effects on breathing plasticity agrees with a wide variety of findings showing that early inflammation induces long-lasting effects on a several brain circuits (Walker et al., 2003; Spencer et al., 2006; Li et al., 2010; Kentner et al., 2010; Kirsten et al., 2011; Fan et al., 2011; Wang et al., 2011; Cai et al., 2013; Wang et al., 2013; Rideau Batista Novais et al., 2014; Lan et al., 2015; Comim et al., 2016; Patil et al., 2016; Onufriev et al., 2017; Singh et al., 2017; Réus et al., 2017; including the spinal cord; Walker et al., 2003; Li et al., 2010) and their plasticity (Walker et al., 2003; Rideau Batista Novais et al., 2014; Onufriev et al., 2017). Surprisingly, in contrast with other reports, the effects observed in this study did not correlate with changes in glia morphology (for contrast see: Fan et al., 2011; Wang et al., 2011; Cai et al., 2013; Wang et al., 2013; Yin et al., 2013; Pang et al., 2013; Lan et al., 2015) and the expression of pro-inflammatory genes (for contrast see: Cai et al., 2013; Yin et al., 2013; Patil et al., 2016, Onufriev et al., 2017) or proinflammatory mediators (for contrast see: Spencer et al., 2006; Kentner et al., 2010; Fan et al., 2011; Wang et al., 2013; Cai et al., 2013; Pang et al., 2013; Lunardelli et al., 2015; Lan et al., 2015; Patil et al., 2016; Onufriev et al., 2017; Réus et al., 2017). I consider the findings to be relevant, but also partially similar to the already described changes in other circuits of the brain by the same experimental manipulation (Walker et al., 2003;Spencer et al., 2006; Li et al., 2010; Kentner et al., 2010; Kirsten et al., 2011; Fan et al., 2011; Wang et al., 2011; Cai et al., 2013; Wang et al., 2013; Rideau Batista Novais et al., 2014; Lan et al., 2015; Comim et al., 2016; Patil et al., 2016; Onufriev et al., 2017; Singh et al., 2017; Réus et al., 2017). On the other hand, the mechanisms behind the findings of the present study are still foggy, since it is not clear what is the source of the inflammatory and the non-inflammatory signals involved in the described effects. Considering the contradictory evidence already available (Walker et al., 2003;Spencer et al., 2006; Li et al., 2010; Kentner et al., 2010; Kirsten et al., 2011; Fan et al., 2011; Wang et al., 2011; Cai et al., 2013; Wang et al., 2013; Rideau Batista Novais et al., 2014; Lan et al., 2015; Comim et al., 2016; Patil et al., 2016; Onufriev et al., 2017; Singh et al., 2017; Réus et al., 2017), the clarification of the mechanisms involved in the modulation described in this work would contribute to the relevance of the study.

We thank the reviewer for their comments and the description of our work in the context of the neuroscience field. We agree that aspects of our results are similar to results in other CNS circuits, however, this is the first demonstration of such changes in a vital homeostatic circuit. Neural circuitry involved in respiratory control must adapt continuously to maintain homeostatic conditions, and plasticity is thought to play a key role. Failure of plasticity in this circuitry could have dire consequences, perhaps even explaining the increased mortality in males after neonatal LPS. While we are not making such broad speculations here, our results are novel and distinct in three ways:

1) Unlike much literature on early life events, we demonstrate profound lasting effects after just one neonatal stimulus. Other studies demonstrate alterations in adulthood after a second, adult stimulus (Bilbo and Schwarz, 2009; Wang et al., 2010; Bilbo, 2010; Kirsten et al., 2010, 2011; Roughton et al., 2013).

2) This is the first demonstration of abolition of S-pathway induced respiratory plasticity, which has been proposed to act as a “back-up pathway” to preserve plasticity in the face of inflammation. Thus, we hypothesize that early life inflammatory events represent the most significant perturbation of the respiratory plasticity. Further, we now provide additional mechanistic details about S-pathway abolishment by neonatal inflammation, and suggest astrocyte signaling is permanently impaired after neonatal inflammation (see response to #3).

3) While our data are consistent with increased male vulnerability early in life, we are the first to characterize female responses. Assessment of respiratory control in females and female respiratory motor plasticity fills a major gap in knowledge and directly addresses the NIH research priority for sex differences. The role of sex hormones in respiratory control remains in its infancy (Behan and Kinkead, 2011). Critical periods for sex hormone action exist during development; high testosterone in young males, and fluctuations of estrogen and progesterone in post-pubescent females are normal changes in hormones during development. Yet interactions between an environmental stimulus (like inflammation) and sex/hormonal differences during these critical periods are poorly understood. Neonatal maternal separation elicits a stress response with sexspecific effects and lasting consequences on the hypothalamic-pituitary-adrenal axis, baseline breathing, and chemosensivity, with males showing greater susceptibility (Genest et al., 2004).

Yet despite the longstanding “Barker Hypothesis” (developmental origins of adult disease) (Barker and Osmond, 1986; Barker, 2002) and other calls to explore the effects of neonatal events on respiratory control (Behan and Kinkead, 2011), our studies represent the first step to fill this gap in knowledge.

Determining how neonatal inflammation impairs the respiratory system will elucidate links between neonatal conditions and adult respiratory insufficiency, leading to better treatments to promote breathing at all age groups. Our study has implications for understanding these long-term deficiencies and will facilitate identification of location(s) within the complex respiratory neural circuitry this deficiency occurs. Previous studies (Rosen et al., 2003; Hibbs et al., 2008; Raynes-Greenow et al., 2012) were correlative, and provided important additions to the scientific premise for the present work; but this is the first to directly test the involvement of neonatal inflammation on respiratory circuitry and specific cell types in the CNS.

[Editors’ note: what now follows is the decision letter after the authors submitted for further consideration.]

Summary:The manuscript by Hocker et al., has important implications. Many babies are exposed to inflammatory events early during their development, and due to an improved healthcare, these babies mostly survive and recover. Yet, it is unclear whether this recovery is complete, and whether even in the absence of any acute inflammation there are long-term consequences on respiratory control. The present study addresses this important issue employing an established rodent model that uses respiratory plasticity in the phrenic nerve as a well-defined respiratory output.This manuscript is a resubmission of a previously reviewed manuscript, and the authors performed additional experiments to address several of the reviewers' original comments. Their new experiments shed new light on the potential involvement of astrocytes (as it relates to the S-pathway in case of severe AIH). The authors also added new experimental data indicating distinct differences between the impairment of Q and S-pathways caused by neonatal inflammation. These are major revisions to the previous manuscript – which is particularly remarkable since such developmental studies consume considerable time.The authors also added a discussion of the importance of acute intermittent hypoxia in the context of spinal cord injury and the therapeutic application for motor neuron disease. The authors emphasize the important implication in particular that the application of AIH could be undermined by early inflammatory experiences.The authors also included a table from a separate study that describes in detail the inflammatory gene data that are associated with the LPS induced inflammation. However, it makes sense that these data are not included in the present study – because they are published elsewhere. However, these data were helpful for evaluating the findings of the present study, because their careful characterization confirms that there is impairment of respiratory activity even though the inflammatory gene responses have returned to baseline.This leads to the important conclusion that the observed loss of plasticity is unlikely due to an ongoing inflammation.Essential revisions:The reviewers concluded that no additional experiments will be required. However, some concerns remained unaddressed, and need to be carefully addressed in the revision.1) The authors should include a limitation section that discusses the experimental approach: in particular whether their findings can be translated to a fully integrated/ intact model animal or human. Regardless of how the model might allow one to dissect out pathways that underlie pLTF, if the whole purpose of the reflex is to maintain ventilation to compensate for hypoxia, then it is still important to demonstrate that the findings are relevant to ventilation in intact, unanesthetized animals. Please address what the findings mean behaviorally, if they are transmitted to the muscles of respiration?

The authors thank the reviewers for their comments. We have now included a limitation section in the Discussion section addressing the potential impact to adult animals and humans. While we are still understanding the exact physiological role of respiratory plasticity, the ability for the respiratory system to adapt and learn is critical for maintaining homeostasis. Our results suggest at least two pathways associated with learning in the respiratory system are impaired following just one bout of early life inflammation. This suggests the respiratory system is likely more vulnerable in adulthood. We have added a more thorough discussion of the physiological implications and limitations of our findings:

“Our experimental approach assessed phrenic nerve output in anesthetized rats and may not be generalizable to respiratory control in freely behaving animals or to other forms of motor plasticity. In humans, AIH induces long-term facilitation of ventilation (Mateika and Komnenov, 2017) and strengthens corticospinal pathways to non-respiratory motor-neurons (Christiansen et al., 2018), suggesting our results likely have relevance to mechanisms of human spinal motor plasticity after AIH. While AIH-induced respiratory motor plasticity does not necessarily alter normal homeostatic control of ventilation, the general facilitation of spinal motor output has significant therapeutic potential for treating patients with respiratory and non-respiratory motor limitations (Trumbower et al., 2012, 2017; Nichols et al., 2013; Hayes et al., 2014).”

2) We suggest reducing the number of figures with negative findings, and instead include a table showing what is different between the groups, which pathway is implicated, and the gender effects. The table would allow the reader to get a better conceptual view of all the many variables that are being assessed in this study.

To improve the clarity of our results, we have reorganized and consolidated figures. As we did not find sex differences in Q-pathway or S-pathway evoked motor plasticity, and the only sex difference in response to neonatal inflammation was the acute neonatal male mortality, we feel a table is not necessary. We have removed two figures and reorganized the Results section to further streamline the manuscript. Figure 9 and Figure 10 showing no changes in preBötC and cervical spinal glial morphology have been combined into one figure (now Figure 8). The previous Figure 8, showing phrenic nerve responses to hypoxia, has been converted to a table (Table 1) and moved to the end of the Results section. Additionally, we have combined male and female inflammatory gene expression data since no sex differences in respiratory plasticity were evident and to reduce the number of figures showing negative data.

3) The Abstract is very short and not very informative. The age of the animal at the time of the exposure and the age of the animals when the pLTF was assessed is not included. Moreover, there are sex differences that also are not specified in the Abstract.

The authors thank the reviewers for their comments. We have revised the Abstract to include additional information; however, we are confined by the *eLife* limit of 150 words.

No sex differences were found in any primary outcomes mentioned in the abstract. The sex-differences in neonatal mortality after acute LPS exposure is a relevant finding and may importantly relate to male-specific human mortality; however, the abstract length limitation does not allow adequate discussion of this finding. Our revised Abstract is below.

“Neonatal inflammation is common and has lasting consequences for adult health. We investigated the lasting effects of a single bout of neonatal inflammation on adult respiratory control in the form of respiratory motor plasticity induced by acute intermittent hypoxia, which likely compensates and stabilizes breathing during injury or disease and has significant therapeutic potential. Lipopolysaccharide-induced inflammation at postnatal day four induced lasting impairments in two distinct pathways to adult respiratory plasticity in male and female rats. Despite a lack of adult pro-inflammatory gene expression or alterations in glial morphology, one mechanistic pathway to plasticity was restored by acute, adult anti-inflammatory treatment, suggesting ongoing inflammatory signaling after neonatal inflammation. An alternative pathway to plasticity was not restored by anti-inflammatory treatment, but was evoked by exogenous adenosine receptor agonism, suggesting upstream impairment, likely astrocytic-dependent. Thus, the respiratory control network is vulnerable to early-life inflammation, limiting respiratory compensation to adult disease or injury.”

4) In their response, the authors have clearly indicated the novelty of their relevant findings, in contrast to the wealth of evidence regarding the long-term consequences of early-life inflammation. Which, by the way, was clear in the original submission. However, we think that it is still necessary to carefully deal with the fact that among those published effects there is a long-lasting change in astrocytes and microglia as well as in their inflammatory mediators. Since these changes were not observed in the submitted study, the authors should include a proper discussion of the source of these differences and they should discuss the alternative mechanisms that could potentially be involved. The authors have shown several aspects that are not involved in their novel findings, which we agree is very important. The authors have also included a set of experiments that indirectly indicate that the astrocytic function could be affected by early-life inflammation, but this needs to be carefully discussed. We agree with the authors that although indirect, these experiments hint that astrocytes could have been changed by early-life inflammation. What we would like to see in the revision is that the authors discuss future experiments or avenues that could more directly address their observations.

The authors thank the reviewers for their comments and interest in our study. Similar to other findings, we found no changes in adult inflammatory gene expression after neonatal LPS-induced inflammation. However, a few reports (Boisse et al., 2005; Kentner et al., 2010) have identified region and sex specific changes in adult CNS inflammation, which have been added to the Discussion section. Other perinatal stimuli with associated inflammatory signaling, such as maternal care and maternal diet, do have lasting programming effects, but are more complex stimuli than the neonatal inflammation in this study. Neonatal inflammation alone does prime glial responses to adult stimuli (Burke et al., 2016), but does not alter baseline inflammatory gene expression, consistent with the findings presented here. We have added additional detail on this to the Discussion section. Further, we also now include mention of our ongoing studies examining cell-specific changes to adult microglia and astrocytes, which we hypothesize will ultimately identify how neonatal inflammation impairs adult, respiratory plasticity (Discussion section).